# Comparative and Functional Analyses Reveal Conserved and Variable Regulatory Systems That Control Lasalocid Biosynthesis in Different *Streptomyces* Species

Minghao Liu,[a] Kairui Wang,[a,b] Junhong Wei,[a,b,c] Ning Liu,[a] Guoqing Niu,[a,d] Huarong Tan,[a,b] Ying Huang[a,b]

aState Key Laboratory of Microbial Resources, Institute of Microbiology, Chinese Academy of Sciences, Beijing, China
bCollege of Life Sciences, University of Chinese Academy of Sciences, Beijing, China
cState Key Laboratory of Silkworm Genome Biology, Southwest University, Chongqing, China
dBiotechnology Research Center, Southwest University, Chongqing, China

Minghao Liu, Kairui Wang, and Junhong Wei contributed equally to this work. Author order was determined alphabetically by their surname.

**ABSTRACT** Lasalocid, a representative polyether ionophore, has been successfully applied in veterinary medicine and animal husbandry and also displays promising potential for cancer therapy. Nevertheless, the regulatory system governing lasalocid biosynthesis remains obscure. Here, we identified two conserved (*lodR2* and *lodR3*) and one variable (*lodR1*, found only in *Streptomyces* sp. strain FXJ1.172) putative regulatory genes through a comparison of the lasalocid biosynthetic gene cluster (*lod*) from *Streptomyces* sp. FXJ1.172 with those (*las* and *lsd*) from *Streptomyces lasalocidi*. Gene disruption experiments demonstrated that both *lodR1* and *lodR3* positively regulate lasalocid biosynthesis in *Streptomyces* sp. FXJ1.172, while *lodR2* plays a negative regulatory role. To unravel the regulatory mechanism, transcriptional analysis and electrophoretic mobility shift assays (EMSAs) along with footprinting experiments were performed. The results revealed that LodR1 and LodR2 could bind to the intergenic regions of *lodR1–lodAB* and *lodR2–lodED*, respectively, thereby repressing the transcription of the *lodAB* and *lodED* operons, respectively. The repression of *lodAB–lodC* by LodR1 likely boosts lasalocid biosynthesis. Furthermore, LodR2 and LodE constitute a repressor-activator system that senses changes in intracellular lasalocid concentrations and coordinates its biosynthesis. LodR3 could directly activate the transcription of key structural genes. Comparative and parallel functional analyses of the homologous genes in *S. lasalocidi* ATCC 31180[T] confirmed the conserved roles of *lodR2*, *lodE*, and *lodR3* in controlling lasalocid biosynthesis. Intriguingly, the variable gene locus *lodR1–lodC* from *Streptomyces* sp. FXJ1.172 seems functionally conserved when introduced into *S. lasalocidi* ATCC 31180[T]. Overall, our findings demonstrate that lasalocid biosynthesis is tightly controlled by both conserved and variable regulators, providing valuable guidance for further improving lasalocid production.

**IMPORTANCE** Compared to its elaborated biosynthetic pathway, the regulation of lasalocid biosynthesis remains obscure. Here, we characterize the roles of regulatory genes in lasalocid biosynthetic gene clusters of two distinct *Streptomyces* species and identify a conserved repressor-activator system, LodR2–LodE, which could sense changes in the concentration of lasalocid and coordinate its biosynthesis with self-resistance. Furthermore, in parallel, we verify that the regulatory system identified in a new *Streptomyces* isolate is valid in the industrial lasalocid producer and thus applicable for the construction of high-yield strains. These findings deepen our understanding of regulatory mechanisms involved in the production of polyether ionophores and provide novel clues for the rational design of industrial strains for scaled-up production.

Address correspondence to Ying Huang, huangy@im.ac.cn.

The authors declare no conflict of interest.

**KEYWORDS** lasalocid, biosynthesis regulation, repressor-activator system, regulatory diversification, *Streptomyces*, biosynthetic gene cluster

Actinobacteria, as the most abundant producers of secondary metabolites, produce a variety of complex polyketides with broad-spectrum biological activities (1). Among them, polyether ionophores are a unique class of type I polyketides featuring distinctive tetrahydrofuran and tetrahydropyran rings and are produced exclusively by actinobacteria. These polyketides have been successfully used in veterinary medicine and animal husbandry for more than half a century. Furthermore, they also display promising biological properties, including antiviral, antifungal, anticancer, and anti-inflammatory activities (2). The intriguing activities of ionophore agents can be attributed to their ability to chelate metal cations to form lipid-soluble complexes. These complexes facilitate cation transport across plasma membranes, resulting in membrane depolarization and subsequent cell damage or death (2, 3). Lasalocid (4), one of the seven currently marketed carboxylic polyether ionophores, has been validated as a classical anticoccidial drug for poultry and a growth promoter in ruminants (5, 6). Lasalocid selectively inhibits intracellular sporozoites while remaining relatively noninjurious to the host cell, and it has biased antimicrobial activity against Gram-positive bacteria, whereas most Gram-negative strains are still refractory (7). Recently, this ionophore has been verified as a novel inhibitor that protects cells from multiple deadly bacterial toxins via corrupting vesicular trafficking and Golgi stack homeostasis (8). Moreover, lasalocid and its synthetic analogs have shown high antiproliferative activity against cancer cell lines and, simultaneously, lower cytotoxicity toward nontumor cells than cisplatin, which hints at their potential to be developed as candidates for cancer treatment (2, 3, 9, 10).

Given the broad-spectrum bioactivities and promising applications of polyether ionophores, increasing numbers of studies have been performed on the regulation of their biosynthesis. The biosynthetic gene clusters (BGCs) of nigericin (*nig*) and salinomycin (*sln*) both contain a single pathway-specific regulatory gene, and the overexpression of these genes greatly enhanced the production of the corresponding polyether ionophores (11, 12). Unlike *nig* and *sln*, the BGCs of monensin (*mon*) and nanchangmycin (*nan*) each contain multiple regulatory genes that encode regulators belonging to diverse families, including LAL (large ATP-binding regulators of the LuxR family), SARP (*Streptomyces* antibiotic regulatory proteins), TetR, and AraC. These regulators form complex coregulatory cascades governing the biosynthesis of monensin and nanchangmycin (13, 14). In addition to the above-mentioned cluster-situated regulators (CSRs), the global transcriptional regulator DasR has also been proven to increase monensin production in *Streptomyces cinnamonensis* (15). However, although the biosynthetic pathway for lasalocid has been well elucidated (16), the regulatory mechanism in lasalocid production remains obscure, which hinders its titer improvement and further industrialization. Meanwhile, comparative genomics and phylogenetic analyses have revealed that even BGCs encoding the same molecule may vary in both gene content and organization (17–19). These homologous BGCs normally keep conserved biosynthetic genes and relatively variable regulatory genes, the latter of which promote BGC diversification and can serve as evolutionary evidence (20, 21). Several BGCs for polyether ionophores have been identified, while little is known about their regulatory diversification. Comparative dissection of their regulation systems may provide valuable insights into the evolution of these BGCs.

To date, lasalocid BGCs have been cloned from two closely related strains of *Streptomyces lasalocidi*, ATCC 31180$^T$ (*las*) and ATCC 35851 (*lsd*) (22, 23), and were recently discovered from a potential novel species represented by red soil-derived *Streptomyces* sp. FXJ1.172 (*lod*) (24–26). Here, to elucidate the pathway-specific regulatory mechanism in lasalocid biosynthesis, we conducted comparative and parallel functional analyses of the regulatory genes from different lasalocid BGCs. Compared to *las* and *lsd*, the *lod* cluster contains an exclusive regulatory gene, *lodR1*, in addition to the conserved *lodR2* and *lodR3* genes that are universal in the reported lasalocid BGCs. We reveal that both the variable

and conserved regulatory genes are functional in different lasalocid BGCs, providing evidence for regulatory diversification. We also identified a conserved repressor-activator system composed of LodR2/Las3/Lsd6–7 (unless otherwise specified, the dash indicates a range) and LodE/Las2/Lsd5, which is important for the control of lasalocid concentrations at appropriate levels in the producers.

## RESULTS

**Comparison of the lasalocid BGCs from *Streptomyces*.** In our previous study, we identified a 76,167-bp putative lasalocid gene cluster (*lod* [GenBank accession number KT591188]) from the genome sequence of *Streptomyces* sp. FXJ1.172 (26). Bioinformatics analysis revealed that the cluster contains 22 open reading frames (ORFs), designated *lodA–lodT*. In this study, to unravel the regulatory mechanism of lasalocid biosynthesis in different *Streptomyces* species, we first compared *lod* with the other two lasalocid gene clusters (*las* and *lsd* from *S. lasalocidi* strains). The results showed that all three BGCs share highly conserved biosynthetic genes and two regulatory genes (*lodR2* and *lodR3*) that encode MarR and LuxR family regulators, respectively (Table 1 and Fig. 1). Intriguingly, *lod* contains four exclusive ORFs: *lodR1*, encoding a TetR family regulator; *lodA*, encoding a protein with domains of unknown functions; *lodB*, encoding a monooxygenase; and *lodC*, encoding a 3-oxoacyl-ACP (acyl carrier protein) synthase that is much longer than its putative homologs encoded by *orf09* in *las* and *lsd3* in *lsd* (Table 1 and Fig. 1). This result illustrates the diversity of lasalocid BGCs in *Streptomyces*, notably the diversity of their regulatory elements.

**All three regulatory genes in *lod* function in controlling lasalocid biosynthesis in *Streptomyces* sp. FXJ1.172.** Three genes of the *lod* cluster are proposed to play a regulatory role in lasalocid biosynthesis in *Streptomyces* sp. FXJ1.172, including a variable gene (*lodR1*) and two conserved ones (*lodR2* and *lodR3*). To verify the functions of these genes, we constructed their mutants by disruptive in-frame deletions in the *Streptomyces* sp. FXJ1.172 wild type (FXJ1.172-WT) and designated them Δ*lodR1*, Δ*lodR2*, and Δ*lodR3*, respectively. High-performance liquid chromatography (HPLC) analysis of the culture extracts from these mutants revealed a substantial loss of lasalocid production in the Δ*lodR1* mutant and a complete loss in the Δ*lodR3* mutant but a moderate increase in the Δ*lodR2* mutant (Fig. 2). Complementation with *lodR1*, *lodR2*, and *lodR3* (inserted into pSET152 and driven by $P_{kasO^*}$) in the corresponding mutants partially or completely restored lasalocid production (Fig. 2). These results indicate that *lodR1* and *lodR3* positively regulate lasalocid biosynthesis in *Streptomyces* sp. FXJ1.172, while *lodR2* plays a negative regulatory role.

**LodR1 positively regulates lasalocid biosynthesis by repressing the expression of adjacent *lodA–lodC*, and this variable system of *Streptomyces* sp. FXJ1.172 seems to act similarly when introduced into *S. lasalocidi*.** The variable gene *lodR1* encodes a transcriptional regulator belonging to the TetR family, members of which are involved in various cellular activities, including but not limited to antibiotic production, quorum sensing, and the export of secondary metabolites (27, 28). To unveil the regulatory mechanism of LodR1, we performed transcriptional analysis, electrophoretic mobility shift assays (EMSAs), and DNase I footprinting experiments. Cotranscriptional analysis of *lod* by reverse transcription-PCR (RT-PCR) revealed the existence of at least five transcriptional units (*lodAB*, *lodED*, *lodR2–F*, *lodH–P*, and *lodST*) and four individually transcribed genes (*lodR1*, *lodC*, *lodG*, and *lodQ*) (Fig. 3A). It is noteworthy that *lodR2* was cotranscribed with *lodR3* and *lodF* in one operon. Next, quantitative real-time RT-PCR (RT-qPCR) was conducted to investigate the effect of *lodR1* disruption on lasalocid biosynthesis at the transcriptional level. Total RNAs of FXJ1.172-WT and the Δ*lodR1* mutant were isolated at days 3 to 5. These time points were selected because lasalocid was initially detected in FXJ1.172-WT on the third day of fermentation, followed by a substantial increase in production over the next 2 days (see Fig. S1 in the supplemental material). Compared to that in FXJ1.172-WT, the transcription of *lodA–lodC* was greatly enhanced in the Δ*lodR1* mutant (Fig. 3B), implying that LodR1 negatively regulates *lodA–lodC*. In contrast, the transcript levels of the structural genes (*lodF*, *lodG*, *lodH–P*, and *lodST*) were

**TABLE 1** Deduced functions of genes in the lasalocid biosynthetic gene cluster from *Streptomyces* sp. FXJ1.172

| Gene | Product size (aa) | Homolog in *las* (% identity/% similarity) | Homolog in *lsd* (% identity/ % similarity) | GenBank accession no. of protein homolog (% identity/% similarity); origin | Proposed function |
|---|---|---|---|---|---|
| *lodR1* | 192 | | | WP_076090794.1 (97/97); *Streptomyces* sp. strain IMTB 2501 | TetR family regulator |
| *lodA* | 148 | | | WP_060890961.1 (84/94); *Streptomyces europaeiscabiei* | DUF1772 domain-containing protein |
| *lodB* | 116 | | | WP_100602108.1 (94/99); *Streptomyces* sp. strain CB02959 | Antibiotic biosynthesis monooxygenase |
| *lodC*[a] | 343 | *orf09* (84/91) | *lsd3* (85/91) | WP_057602612.1 (80/84); *Streptomyces* sp. strain Root1310 | 3-Oxoacyl-ACP synthase |
| *lodD* | 245 | *las1* (83/90) | *lsd4* (83/90) | CAQ64680.1 (83/90); *Streptomyces lasalocidi* | Type II thioesterase |
| *lodE* | 522 | *las2* (87/92) | *lsd5* (87/92) | BAG85020.1 (87/92); *Streptomyces lasalocidi* | Putative resistance protein |
| *lodR2* | 161 | *las3* (93/95) | *lsd6–7* (93/95) | CAQ64682.1 (93/95); *Streptomyces lasalocidi* | MarR family regulator |
| *lodR3* | 884 | *las4* (80/86) | *lsd8* (80/86) | CAQ64683.1 (80/86); *Streptomyces lasalocidi* | LuxR family regulator |
| *lodF* | 447 | *las5* (91/94) | *lsd9* (92/95) | CAQ64684.1 (91/94); *Streptomyces lasalocidi* | Crotonyl-CoA reductase |
| *lodG* | 572 | *las6* (85/90) | *lsd10* (85/90) | BAG85025.1 (85/90); *Streptomyces lasalocidi* | 3-Hydroxybutyryl-CoA dehydrogenase |
| *lodH* | 1,944 | *lasAI* (80/85) | *lsd11* (80/85) | CAQ64686.1 (80/85); *Streptomyces lasalocidi* | Polyketide synthase |
| *lodI* | 710 | *lasAII* (78/86) | *lsd12* (78/86) | CAQ64687.1 (78/86); *Streptomyces lasalocidi* | Polyketide synthase |
| *lodJ* | 1,115 | *lasAIII* (84/89) | *lsd13* (84/89) | BAG85028.1 (84/89); *Streptomyces lasalocidi* | Polyketide synthase |
| *lodK* | 1,650 | *lasAIV* (82/87) | *lsd14* (82/87) | BAG85029.1 (82/87); *Streptomyces lasalocidi* | Polyketide synthase |
| *lodL* | 3,071 | *lasAV* (72/79) | *lsd15* (72/79) | CAQ64690.1 (72/79); *Streptomyces lasalocidi* | Polyketide synthase |
| *lodM* | 2,909 | *lasAI* (76/82) | *lsd11* (76/82) | BAG85026.1 (76/82); *Streptomyces lasalocidi* | Polyketide synthase |
| *lodN* | 5,437 | *lasAII* (70/77) | *lsd12* (70/77) | CAQ64687.1 (70/77); *Streptomyces lasalocidi* | Polyketide synthase |
| *lodO* | 1,900 | *lasAVI* (84/88) | *lsd16* (83/88) | CAQ64691.1 (84/88); *Streptomyces lasalocidi* | Polyketide synthase |
| *lodP* | 1,309 | *lasAVII* (81/87) | *lsd17* (79/85) | CAQ64692.1 (81/87); *Streptomyces lasalocidi* | Polyketide synthase |
| *lodQ* | 122 | *las7* (93/97) | | CAQ64693.1 (93/97); *Streptomyces lasalocidi* | Hypothetical protein |
| *lodS* | 472 | *lasC* (89/93) | *lsd18* (89/93) | B5M9L6.1 (89/93); *Streptomyces lasalocidi* | Putative epoxidase |
| *lodT* | 275 | *lasB* (85/91) | *lsd19* (85/90) | CAQ64695.1 (85/91); *Streptomyces lasalocidi* | Epoxide hydrolase |

[a]LodC (343 amino acids [aa]) is much longer than its putative homologs ORF09 (141 aa) and Lsd3 (144 aa).

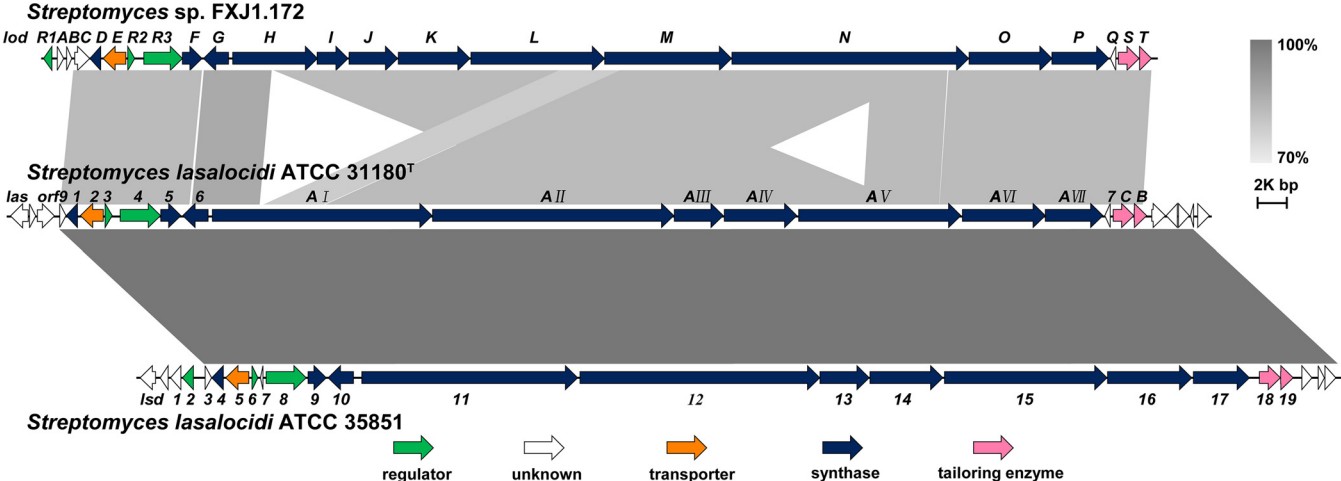

**FIG 1** Comparison of the genetic organizations of different lasalocid biosynthetic gene clusters. The figure was obtained using Easyfig, where grayscale bars represent regions of shared similarity according to BLASTN analysis.

decreased in the Δ*lodR1* mutant (Fig. 3B). To determine the target genes of LodR1, EMSAs were performed with all potential interoperonic promoters indicated in Fig. 3A and the intergenic region of *lodR2–lodR3* (a possible intraoperonic promoter may exist in this region of over 500 bp in length). Recombinant protein LodR1-His$_6$ was purified from *Escherichia coli* C41(DE3) containing expression plasmid pET23b::*lodR1*. The result showed that LodR1 could specifically bind to the intergenic region of *lodR1–lodA* (P$_{R1–A}$) rather than to the other promoter regions in *lod* or the promoter region of *hrdB* (Fig. 4A and Fig. S2). LodR1 comprises an N-terminal DNA-binding domain and a C-terminal ligand-binding domain and shows high identity to SCO6121 from *Streptomyces coelicolor* and SVEN_3912 from *Streptomyces venezuelae* (both with 88% identity). To investigate the potential substrate of the ligand-binding domain in LodR1, EMSAs were performed with P$_{R1–A}$ in the presence of different antibiotics (including lasalocid as the most probable ligand and other antibiotics with structural diversity, such as ampicillin, apramycin, and chloromycetin). Yet no perturbation of the DNA-binding activity of LodR1 was observed (data not shown), indicating that these antibiotics are not ligands of LodR1. To further elucidate the exact DNA-binding sites of LodR1, DNase I footprinting was carried out with probe P$_{R1–A}$. The results showed that LodR1 protected a region from nucleotide (nt) −52 to nt +38 relative to the *lodA* transcription start point (TSP) (Fig. 4B). Analysis of this region identified a conserved recognition sequence (5′-CTAGCGTTGC-3′) forming a palindromic structure near the *lodA* TSP (Fig. 4C). Together, these data suggest that LodR1 directly represses the expression of *lodAB*. As *lodC* was not cotranscribed with *lodAB* and its intergenic region with *lodB* could not be recognized by LodR1, the increase in the *lodC* transcript level in the Δ*lodR1* mutant is likely due to the indirect repression of *lodC* by LodR1.

Although the *lodR1–lodC* locus was found only in *Streptomyces* sp. FXJ1.172, we speculate that it might also function in heterologous lasalocid producers. To test this speculation, the *lodA–lodC* and *lodR1–lodC* loci were heterologously expressed in *S. lasalocidi* ATCC 31180$^T$ to generate strains 31180-KABC and 31180-R1ABC, respectively. As expected, the constitutive expression of *lodA–lodC* resulted in a dramatic decrease in lasalocid production in 31180-KABC (Fig. 4D). In contrast, lasalocid production in 31180-R1ABC was comparable to that in the WT strain, which might be attributed to the inhibition of *lodA–lodC* expression by LodR1. These findings suggest that the variable genes *lodR1* and *lodA–lodC* seem to function conservatively in lasalocid biosynthesis in different *Streptomyces* species.

**LodR2 and LodE constitute a conserved repressor-activator system that senses changes in the concentration of lasalocid and coordinates its biosynthesis.** The regulatory protein encoded by *lodR2* contains DNA-binding domains with a characteristic

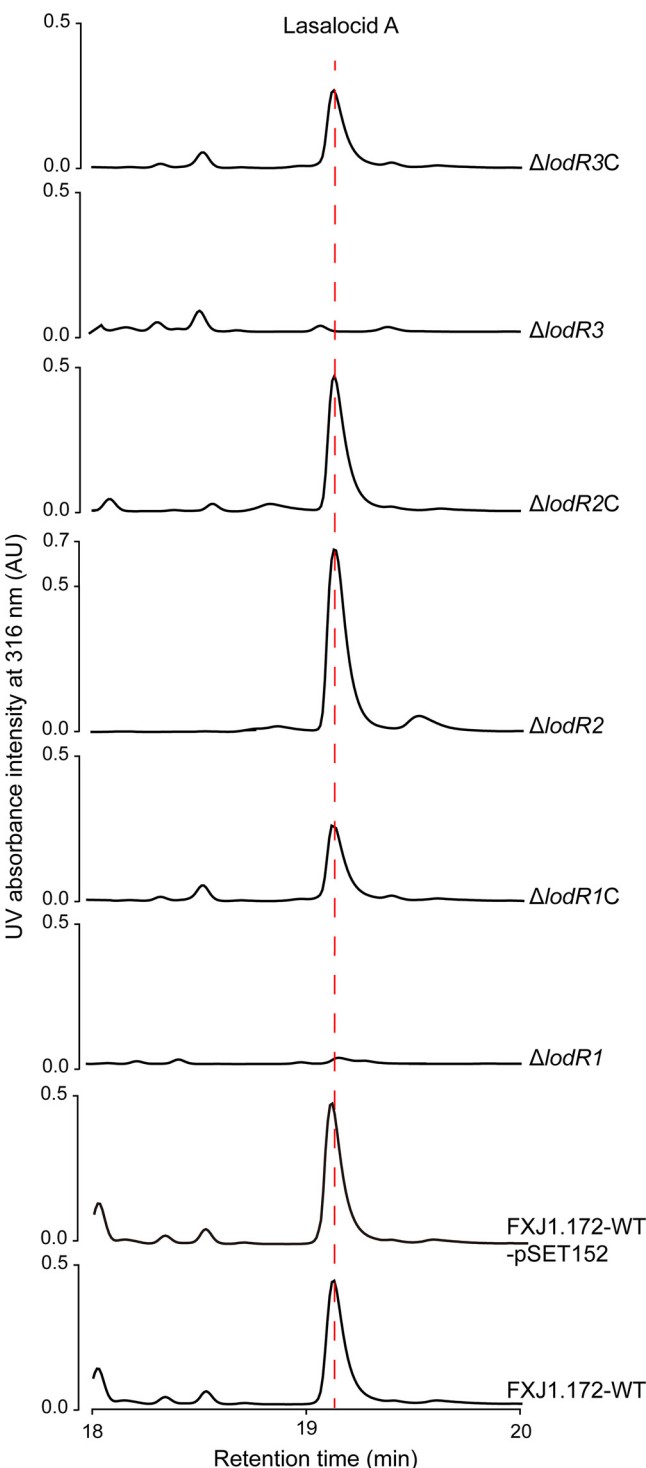

**FIG 2** HPLC analysis of lasalocid A production in the *Streptomyces* sp. FXJ1.172 wild type (FXJ1.172-WT) and its derivatives after fermentation for 240 h. Δ*lodR1*, Δ*lodR2*, and Δ*lodR3* represent *lodR1*, *lodR2*, and *lodR3* mutants, respectively, constructed by disruptive in-frame deletion; Δ*lodR1*C, Δ*lodR2*C, and Δ*lodR3*C represent the corresponding complemented strains of the mutants. AU, arbitrary units.

winged helix-turn-helix fold that is conserved in the MarR family of transcriptional regulators (29). As a common feature of MarR members, these regulators can interact with specific ligands to modulate the DNA-binding affinity, thus regulating the transcription of their cognate genes (30). LodR2 exhibits extremely high identity (93%) to its

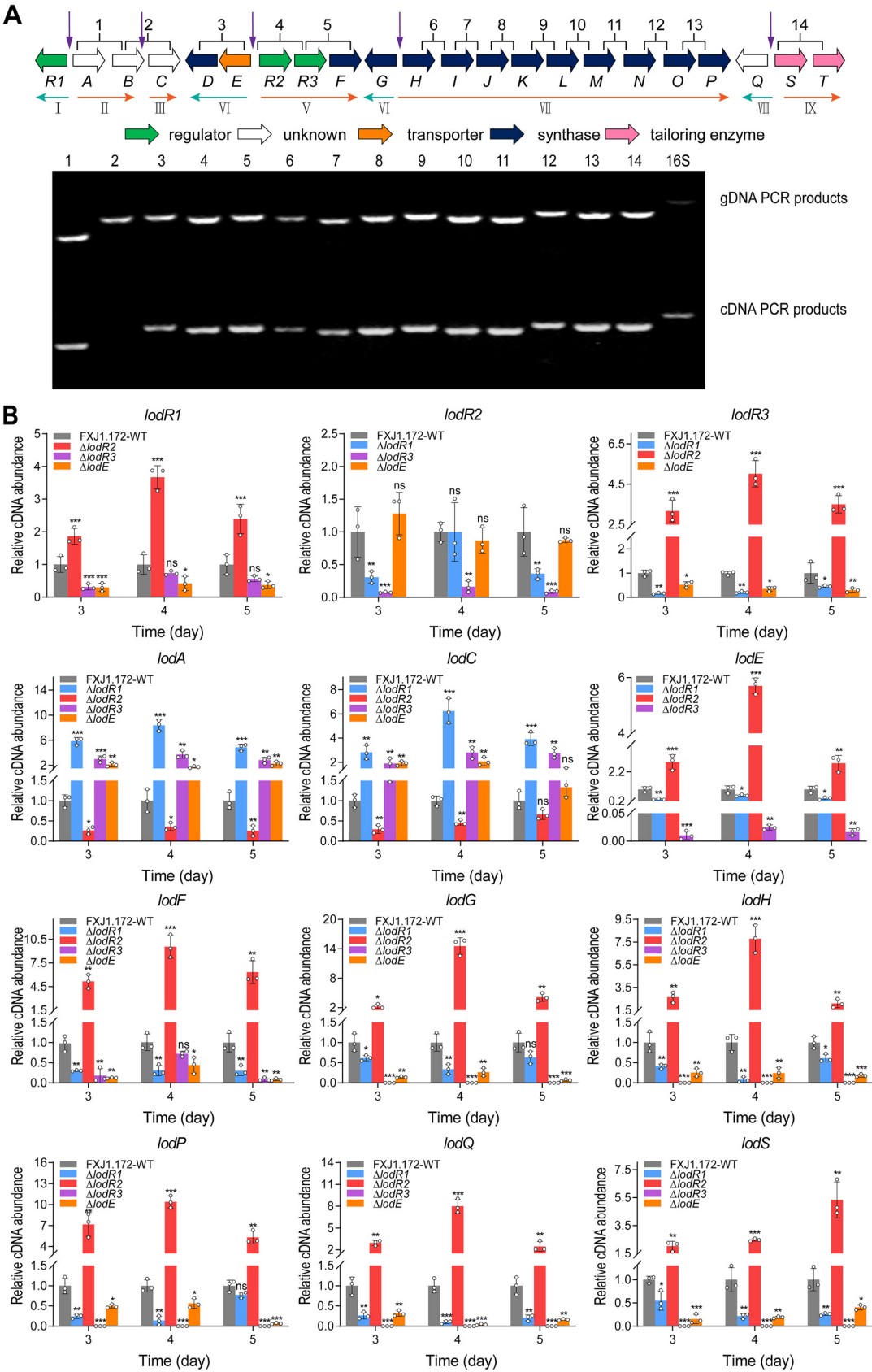

**FIG 3** Transcriptional analysis of the *lod* cluster. (A) Cotranscriptional analysis of the *lod* genes by RT-PCR. The regions used for PCR amplification are labeled 1 to 14; the operons (I to IX) and their transcriptional directions are marked by horizontal

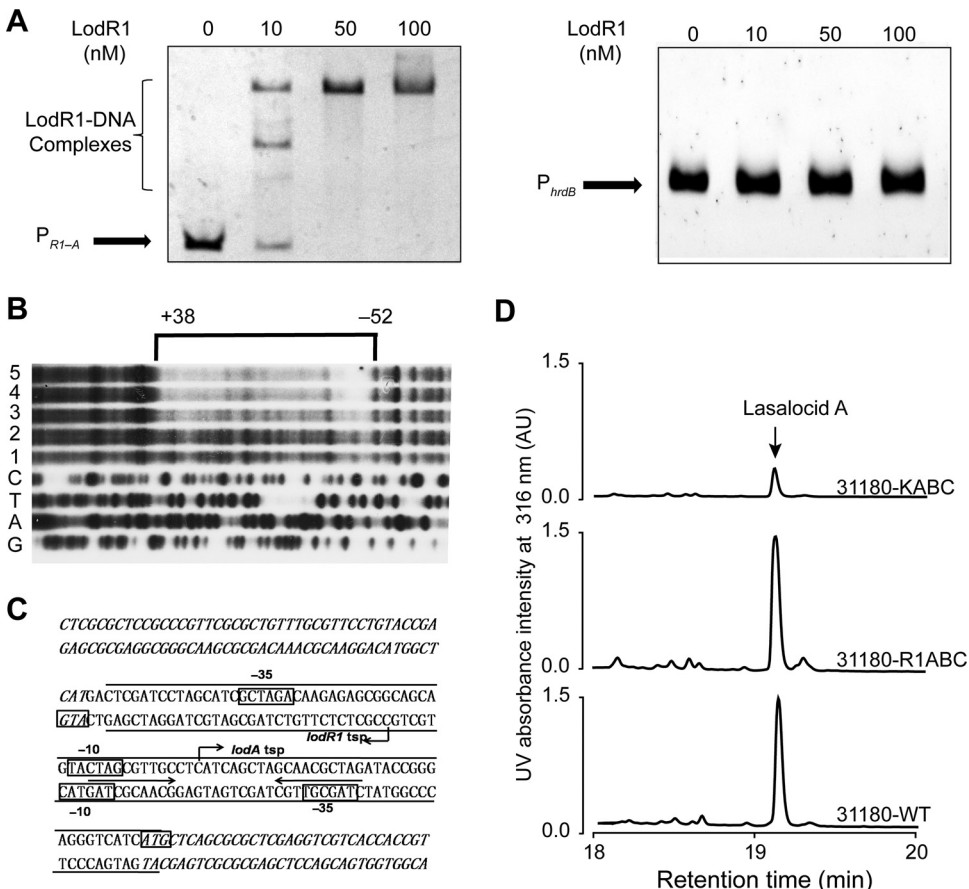

**FIG 4** Functional characterization of the regulator LodR1 in lasalocid biosynthesis. (A) EMSA of LodR1 binding to the intergenic region of *lodR1–lodA*. All potential promoters indicated in Fig. 3A and the intergenic region of *lodR2–lodR3* were used for EMSAs. P$_{R1-A}$ and P$_{hrdB}$ indicate the intergenic region of *lodR1–lodA* and the promoter region of *hrdB*, respectively. (B) Determination of LodR1-binding sites on P$_{R1-A}$ by footprinting experiments. The concentrations of LodR1 used in lanes 1 to 5 were 0, 0.2, 0.4, 0.8, and 1.0 $\mu$M, respectively. The bracket denotes the region protected by LodR1, and the numbers at the top indicate the distances relative to the transcription start point (TSP) of *lodA*. (C) Nucleotide sequences of the *lodR1–lodA* intergenic region and LodR1-binding sites. The predicted TSP is indicated by a bent arrow. The LodR1-binding sites identified in the footprinting experiment are indicated between long lines, and the putative −10 and −35 regions are marked by boxes. The coding regions of *lodA* and *lodR1* are italicized. The inverted repeats are marked by arrows. (D) HPLC analysis of lasalocid A production in the *Streptomyces lasalocidi* ATCC 31180$^T$ wild type (31180-WT) and its derivatives 31180-KABC and 31180-R1ABC after fermentation for 240 h. 31180-KABC contains plasmid pSET152 with locus *lodA–C* driven by P$_{kasO*}$, and 31180-R1ABC contains plasmid pSET152 with locus *lodR1–C*.

homologs Las3 from *S. lasalocidi* ATCC 31180$^T$ and Lsd6–7 (which should be an intact ORF but was likely interrupted and split into two truncated ORFs by misannotation) from *S. lasalocidi* ATCC 35851 (Table 1), suggesting a conserved role of this MarR regulator in lasalocid biosynthesis. To functionally characterize the regulatory role of LodR2, experiments similar to those described above for LodR1 were carried out. RT-qPCR results showed that the transcript levels of *lodR3* and all of the structural genes were significantly increased in the Δ*lodR2* mutant compared to those in FXJ1.172-WT (Fig. 3B). This finding indicates that LodR3 may activate the expression of structural

**FIG 3** Legend (Continued)

arrows. Total RNAs were isolated from FXJ1.172-WT after incubation for 96 h and used for synthesizing cDNA. The genomic DNA (gDNA) was used as a positive control for the template. The 16S rRNA gene was used as a positive control for the genes. All potential interoperonic promoter regions are indicated by vertical arrows. (B) Transcriptional analysis of representative genes in *lod* from the FXJ1.172-WT, Δ*lodR1*, Δ*lodR2*, Δ*lodR3*, and Δ*lodE* strains by RT-qPCR. Total RNAs were isolated from the strains after incubation for 72, 96, and 120 h. The relative transcript level of each gene was normalized against the level of the internal reference gene (16S rRNA) at the corresponding time points. Error bars show the standard deviations from three independent experiments. Data for the mutants were compared to those for FXJ1.172-WT, and statistical significance was determined by Student's *t* test. *, $P < 0.05$; **, $P < 0.01$; ***, $P < 0.001$; ns, not significant.

genes. Meanwhile, the transcription of *lodED* was greatly enhanced in the Δ*lodR2* mutant, indicating that LodR2 negatively regulates *lodED*. EMSAs of LodR2-His$_6$ with all potential interoperonic promoters and the intergenic region of *lodR2*–*lodR3* also proved that LodR2 could specifically bind to the intergenic region of *lodR2*–*lodE* (Fig. 5A and Fig. S2). Footprinting showed that LodR2 protected a region from nt −65 to nt +88 relative to the *lodE* TSP (Fig. 5B). There is a conserved sequence, 5′-CATTCRT-3′ (where R is A or G), in the *lodR2*–*lodE* intergenic region, forming a palindromic structure near the TSP of *lodR2* (Fig. 5C). MarR family regulators, especially the cluster-situated ones, commonly have ligand-binding domains. To test whether the DNA-binding activity of LodR2 is modulated by lasalocid, EMSAs in the presence of different antibiotics were performed. The addition of lasalocid led to the dissociation of LodR2 from P$_{R2–E}$, while the addition of the other antibiotics tested showed no effect on LodR2 (Fig. 5D), indicating that lasalocid interacts specifically with LodR2 and abolishes its DNA-binding activity.

As the Δ*lodR2* mutant exhibited enhanced lasalocid production and LodR2 directly repressed the transcription of *lodED* (Fig. 2 and 3 and Fig. 5A to C), it was suggested that the proteins encoded by *lodED* could activate lasalocid biosynthesis. Based on the proposed functions of *lodE* and *lodD* (Table 1), we speculate that *lodE* (encoding a putative resistance protein) is vital for lasalocid biosynthesis. To test our assumption, a *lodE* mutant (Δ*lodE*) was constructed by disruptive in-frame deletion. Compared to FXJ1.172-WT, the Δ*lodE* mutant produced much less lasalocid (Fig. 5E), which was consistent with the significantly decreased transcription of the structural genes *lodF*–*lodS* in this mutant (Fig. 3B). Sequence alignment indicated that LodE belongs to the major facilitator superfamily of transporters with relatively high protein similarity to MonT (53% identity), a putative efflux protein responsible for self-resistance to monensin in *Streptomyces cinnamonensis* (31). To ascertain whether LodE acts as a transporter of lasalocid, we compared the intracellular and extracellular distributions of lasalocid in FXJ1.172-WT and the Δ*lodE* mutant. As expected, the ratio of intracellular/extracellular lasalocid concentrations of the Δ*lodE* mutant was significantly higher than that of FXJ1.172-WT (Fig. 5F), confirming that LodE is a transporter of lasalocid. Moreover, FXJ1.172-WT and the Δ*lodE* mutant had comparable growth rates and biomasses (Fig. S3), excluding the possible effect of the growth perturbation of the mutant on lasalocid production.

**LodR3 positively regulates the expression of key structural genes in *lod*.** The third regulatory gene, *lodR3*, encodes a prototypical LuxR family transcriptional regulator containing an N-terminal ATPase domain and a putative C-terminal DNA-binding domain. The ATPase-associated LuxR regulators are usually involved in antibiotic biosynthesis in *Streptomyces* (32). LodR3 shows strong sequence identity (80%) to both Las4 from *S. lasalocidi* ATCC 31180$^T$ and Lsd8 from *S. lasalocidi* ATCC 35851 (Table 1). RT-qPCR analysis supported the above-mentioned result that LodR3 could positively regulate lasalocid biosynthesis. Very few transcripts of the structural genes were detected in the Δ*lodR3* mutant compared to FXJ1.172-WT (Fig. 3B), indicating that LodR3 is essential for initiating the transcription of the key structural genes. Unfortunately, all efforts to express soluble LodR3-His$_6$ were unsuccessful. Thus, *gusA* transcriptional fusion assays were performed to determine the potential target genes of LodR3 as previously described (33). According to the results of the cotranscriptional analysis mentioned above (Fig. 3A), the putative promoters of *lodC*, *lodG*, *lodH*, *lodQ*, *lodS*, and the intergenic region of *lodR2*–*lodR3* were tested using *Streptomyces coelicolor* M1146 as a host (Fig. 6A). As a result, blue color was detected in *S. coelicolor* M1146 derivatives containing *lodR3* driven by P$_{kasO*}$ and *gusA* driven by the above-mentioned promoters except for the *lodC* promoter (Fig. 6B). This result indicates that *lodR3* may have its own promoter in the intergenic region of *lodR2*–*lodR3* in addition to cotranscription with *lodR2*. It also suggests that the promoters of *lodR3*, *lodG*, *lodH*, *lodQ*, and *lodS* are the targets of LodR3 and that LodR3 can therefore directly activate the transcription of *lodR3*–*F*, *lodG*, *lodH*–*P*, *lodQ*, and *lodST*.

**Comparative and parallel functional analyses confirmed the conserved roles of *lodE*, *lodR2*, and *lodR3* homologs in controlling lasalocid biosynthesis in *S. lasalocidi*.** *S. lasalocidi* strains are used as industrially valuable lasalocid producers, and thus, the regulatory mechanism of their lasalocid biosynthesis is worth dissecting for the

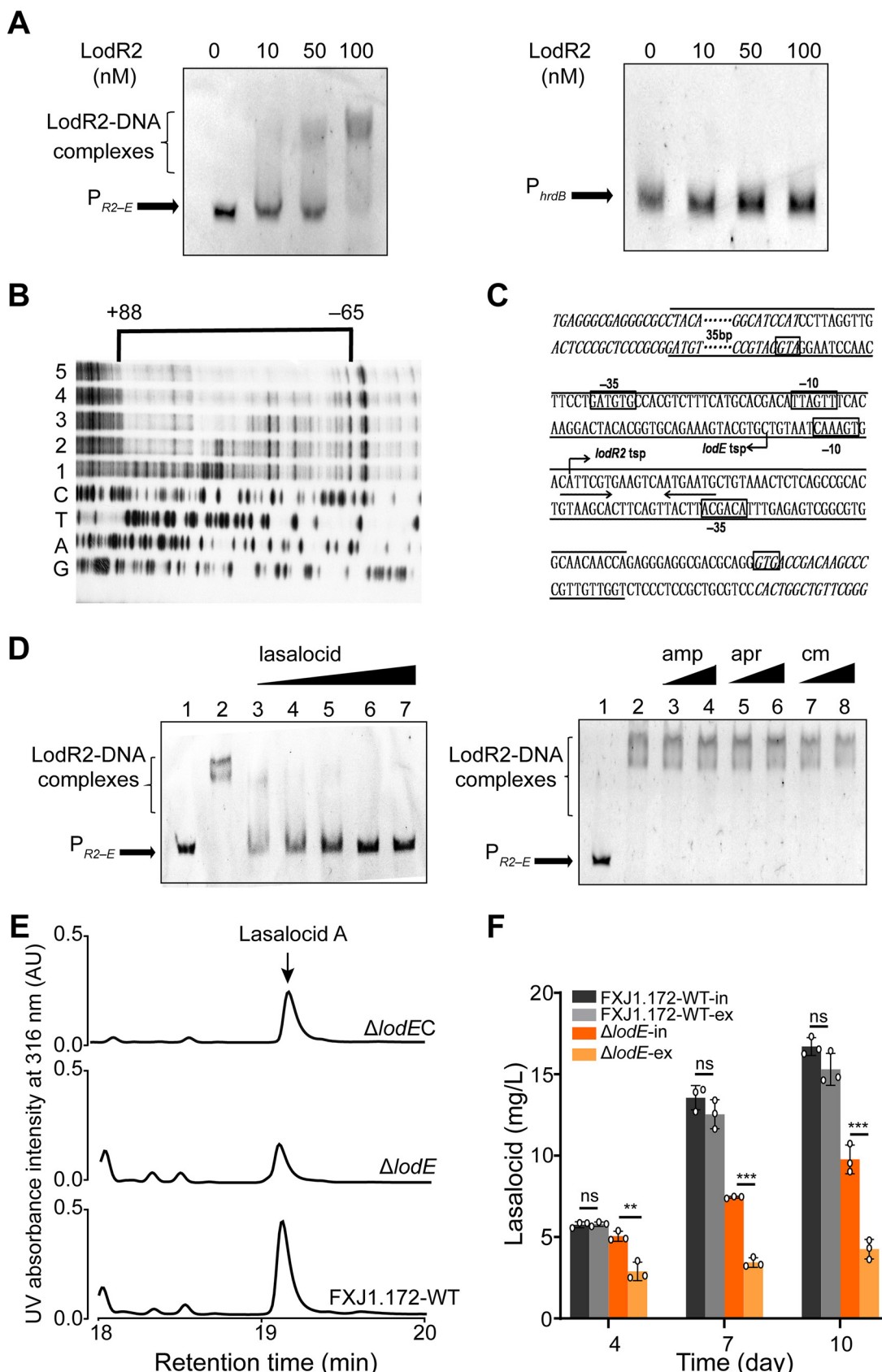

**FIG 5** Functional characterization of the regulator LodR2 in lasalocid biosynthesis. (A) EMSA of LodR2 binding to the intergenic region of *lodE–lodR2*. All potential promoters indicated in Fig. 3A and the intergenic region of *lodR2–lodR3* were

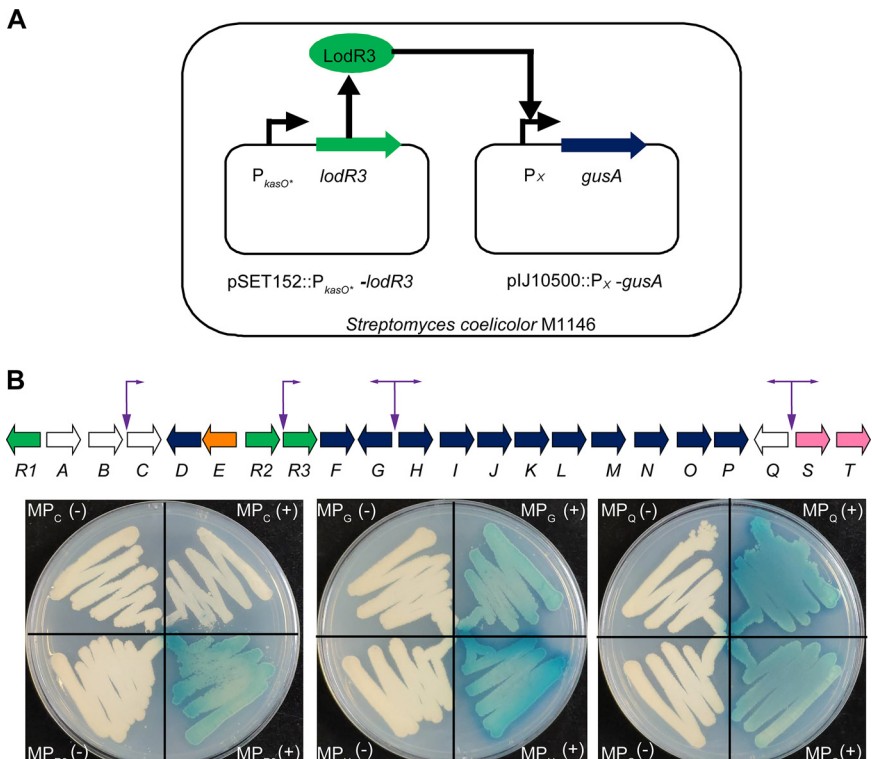

**FIG 6** Determination of the potential target genes of LodR3 by *gusA* transcriptional fusion assays. (A) Schematic diagram of *gusA* transcriptional fusions. Plasmid pSET152::P$_{kasO*}$-*lodR3* was used to overexpress *lodR3*, and the pIJ10500::P$_x$-*gusA* plasmid contains a reporter gene, *gusA*, driven by the promoter (P$_x$) of *lodC*, *lodR3*, *lodG*, *lodH*, *lodQ*, or *lodS*. Both pSET152::P$_{kasO*}$-*lodR3* and pIJ10500::P$_x$-*gusA* plasmids were introduced into *S. coelicolor* M1146 to determine the target genes of LodR3. (B) Chromogenic GUS assays of *S. coelicolor* M1146 derivatives. Potential promoters tested and their directions of transcription initiation are indicated by vertical and horizontal arrows, respectively. MP$_x$(−) strains contain only pIJ10500::P$_x$-*gusA* plasmids, and MP$_x$(+) strains contain both pIJ10500::P$_x$-*gusA* and pSET152::P$_{kasO*}$-*lodR3* plasmids.

construction of high-yield strains. As *las* and *lsd* are very similar clusters from the same species, we then selected *las* (which is from the publicly available *S. lasalocidi* type strain) to compare the functions of its regulatory genes with those in *lod*. Homologs of *lodE*, *lodR2*, and *lodR3* were identified in *las*, which have been designated *las2*, *las3*, and *las4*, respectively (22). To verify the role of these genes in lasalocid biosynthesis, the corresponding disruption mutants (Δ*las2*, Δ*las3*, and Δ*las4*) were constructed from the *S. lasalocidi* ATCC 31180$^T$ wild type (31180-WT). Compared to that of 31180-WT, the lasalocid yield of the Δ*las2* mutant was greatly reduced (from 33.8 to 4.7 mg/L), and that of the Δ*las3* mutant was increased (from 33.8 to 112.5 mg/L), while lasalocid was no longer

**FIG 5** Legend (Continued)

used for EMSAs. P$_{R2−E}$ and P$_{hrdB}$ indicate the intergenic region of *lodR2*–*lodE* and the promoter region of *hrdB*, respectively. (B) Determination of LodR2-binding sites on P$_{R2−E}$ by footprinting experiments. The concentrations of LodR2 used in lanes 1 to 5 were 0, 0.2, 0.4, 0.8, and 1.0 μM, respectively. The bracket denotes the region protected by LodR2, and the numbers at the top indicate the distances relative to the TSP of *lodE*. (C) Nucleotide sequences of the *lodR2*–*lodE* intergenic region and LodR2-binding sites. The predicted TSP is indicated by a bent arrow. The LodR2-binding sequences identified in the footprinting experiment are indicated between long lines, and the putative −10 and −35 regions are marked by boxes. The coding regions of *lodE* and *lodR2* are italicized. The inverted repeats are marked by arrows. (D) Effects of different antibiotics on the DNA-binding activity of LodR2. Each lane contained probe P$_{R2−E}$. Lanes 2 to 7 contained 100 nM LodR2. (Left) Effects of lasalocid. Lanes 3 to 7 contained 0.4, 0.8, 1.6, 3.2, and 6.4 μM lasalocid, respectively. (Right) Effects of ampicillin (amp), apramycin (apr), and chloromycetin (cm). Lanes 3 to 7 contained the antibiotics at a concentration of 3.2 or 6.4 μM. (E) HPLC analysis of lasalocid A production in FXJ1.172-WT and its Δ*lodE* and Δ*lodE*C derivatives after fermentation for 240 h. Δ*lodE*, *lodE*-disruptive in-frame deletion mutant; Δ*lodE*C, *lodE*-complemented Δ*lodE* strain. (F) Intracellular (in) and extracellular (ex) distributions of lasalocid in FXJ1.172-WT and the Δ*lodE* mutant. Error bars show the standard deviations from three independent experiments. Statistical significance was determined by Student's *t* test. *, $P < 0.05$; **, $P < 0.01$; ***, $P < 0.001$; ns, not significant.

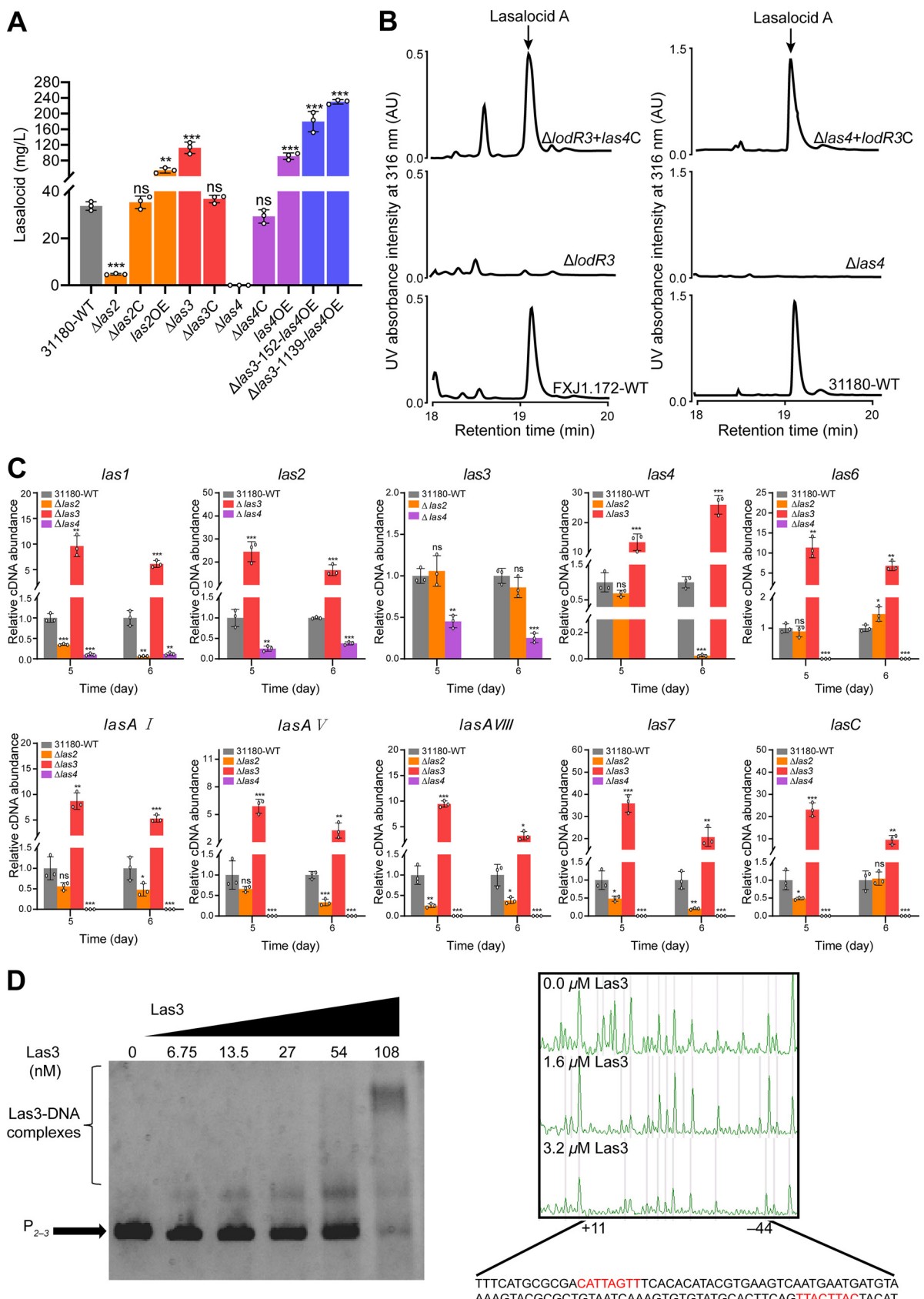

**FIG 7** Parallel functional verification of the *lodE*, *lodR2*, and *lodR3* homologs in *las* from *S. lasalocidi* ATCC 31180[T] and construction of high-yield lasalocid-producing strains. (A) Lasalocid A production in the *S. lasalocidi* ATCC 31180[T] wild type (31180-WT) and its derivatives.

detected in the Δ*las4* mutant (Fig. 7A). Complementation with the genes in the disruption mutants restored the yields (Fig. 7A). To verify the functional conservation of *las4* and its homolog *lodR3*, cross-complementation of the two genes was conducted. As a result, both the Δ*lodR3* mutant cross-complemented with *las4* and the Δ*las4* mutant cross-complemented with *lodR3* could restore lasalocid production to the level of the WT strains (Fig. 7B). Cotranscriptional analysis of *las* revealed the existence of at least four transcriptional units (*las1–2*, *las3–5*, *lasAI–lasAVII*, and *lasCB*) and two individual transcriptional genes (*las6* and *las7*) (Fig. S4). Similar to their homologs in *lod*, *las3–5* form a single operon. Likewise, RT-qPCR analysis of *las* genes in 31180-WT and the disruption mutants displayed transcriptional patterns similar to those observed for *lod* genes in *Streptomyces* sp. FXJ1.172 (Fig. 7C). Moreover, soluble Las3-His$_6$ protein was successfully expressed in *E. coli* BL21(DE3) using the expression plasmid pET28a::*las3*. Next, EMSAs and footprinting assays were performed to determine its target genes and binding sequences. As expected, Las3-His$_6$ formed complexes with the intergenic region of *las2–las3* and protected a region from nt −44 to nt +11 relative to the *las2* TSP (Fig. 7D). Analysis of the binding sites revealed an 8-bp palindromic sequence near the *las2* TSP (consensus, 5′-CATTMRTT-3′ [where M is C or A and R is A or G]) (Fig. 7D), which is similar to the 5′-CATTCRT-3′ (where R is A or G) sequence found in the *lodR2–lodE* intergenic region from *Streptomyces* sp. FXJ1.172. All of these results are consistent with those for *Streptomyces* sp. FXJ1.172, indicating a conserved regulatory mechanism of lasalocid biosynthesis in the two species. Based on the above-described regulatory studies, high-yield lasalocid-producing strains were constructed by overexpressing *las4* in the Δ*las3* mutant via a multicopy vector, pKC1139::P$_{kasO*}$*las4*, or a single-site integrative vector, pSET152::P$_{kasO*}$*las4*. The resultant mutants, Δ*las3*-1139-*las4*OE and Δ*las3*-152-*las4*OE, enhanced their lasalocid yields by 5.1 and 6.8 times, respectively, compared to 31180-WT (Fig. 7A). Overall, by the simultaneous disruption of the negative regulatory gene *las3* and the overexpression of the positive regulatory gene *las4*, we obtained high-yield strains with lasalocid production of up to 230 mg/L.

## DISCUSSION

Multiple regulatory genes coexist in lasalocid BGCs, but they may not form the complex coregulatory cascade process seen in the biosynthesis of monensin and nanchangmycin (13, 14). Our results reveal that lasalocid biosynthesis in different *Streptomyces* species is controlled by one variable and two conserved regulatory systems, providing an understanding of the regulatory diversification and mechanism. To our surprise, no intact homologs of *lodR1–lodC* were identified in the NCBI nucleotide database. The presence of this variable locus in *Streptomyces* sp. FXJ1.172 could be partially ascribed to the divergent evolution of the regulation of lasalocid biosynthesis among streptomycetes. The distant phylogenetic relationship between *Streptomyces* sp. FXJ1.172 and *S. lasalocidi* (96.2% 16S rRNA gene sequence similarity and 81.7% genome-wide average nucleotide identity) (24) and their disparate soil origins, red soil in China (25) and park soil in the United States (34), respectively, both support the divergent-evolution hypothesis. Nevertheless, more lasalocid BGCs are needed to reconstruct an evolutionary roadmap of the biosynthesis regulation of this polyether. The *lsd2* gene, encoding a putative IclR family protein (23), might also have a regulatory

**FIG 7** Legend (Continued)

Δ*las2*, Δ*las3*, and Δ*las4* denote *las2*, *las3*, and *las4* disruption mutants, respectively; Δ*las2*C, Δ*las3*C, and Δ*las4*C denote the corresponding complemented strains of the disruption mutants; and Δ*las3*-152-*las4*OE and Δ*las3*-1139-*las4*OE denote *las4* overexpressed in the Δ*las3* mutant via vector pSET152::P$_{kasO*}$*las4* and vector pKC1139::P$_{kasO*}$*las4*, respectively. (B) HPLC analysis of lasalocid A production in the Δ*lodR3* mutant cross-complemented with *las4* (Δ*lodR3*+*las4*C) and the Δ*las4* mutant cross-complemented with *lodR3* (Δ*las4*+*lodR3*C). (C) Transcriptional analysis of representative genes in *las* from the 31180-WT, Δ*las2*, Δ*las3*, and Δ*las4* strains by RT-qPCR. Total RNAs were isolated from the strains after incubation for 120 and 144 h. The relative transcript levels of each gene were normalized against the level of the internal reference gene (16S rRNA) at the corresponding time points. Error bars show the standard deviations from three independent experiments. Data for the mutants were compared to those for 31180-WT, and statistical significance was determined by Student's *t* test. *, $P < 0.05$; **, $P < 0.01$; ***, $P < 0.001$; ns, not significant. (D) EMSA and footprinting experiment of Las3. P$_{2–3}$ represents the intergenic region of *las2* and *las3*, and the Las3-protected sequence in P$_{2–3}$ is indicated at the bottom of the graph, with the 8-bp palindromic sequence in red.

role in lasalocid biosynthesis in *S. lasalocidi* ATCC 35851, albeit the whole sequence of *lsd* needs to be confirmed by genome sequencing. It was unexpected that the constitutive heterologous expression of *lodA–lodC* from *Streptomyces* sp. FXJ1.172 also resulted in the diminished production of lasalocid in *S. lasalocidi* ATCC 31180[T]. This result suggests that these ORFs exclusive to *lod* can play a general role in the regulation of lasalocid biosynthesis in different *Streptomyces* species. Although we attributed the undisturbed lasalocid production in the mutant strain 31180-R1ABC to the inhibition of *lodA–lodC* expression by LodR1, we cannot exclude another possibility, that all of these genes are not well expressed in the heterologous host, and more details are needed to further check whether the repression of the expression of *lodA–lodC* by LodR1 is functionally conserved in *S. lasalocidi*. Additionally, based on their proposed roles (Table 1), we presumed that *lodA–lodC* might be involved in the biosynthesis of an unknown molecule, which shares substrates with lasalocid biosynthesis, thus leading to the reduced production of lasalocid. However, no additional metabolites were detected in *S. coelicolor* M1146 or *S. lasalocidi* ATCC 31180[T] derivatives containing heterologously expressed *lodA–lodC* (data not shown). Notably, *S. lasalocidi* has only about 1/3 of LodC encoded within its lasalocid BGC, which keeps only a residue of the conserved domain found in LodC and thus looks like a truncated and useless protein (Table 1). Our further work will focus on functional analyses of *lodA*, *lodB*, and *lodC* to determine how much of the reduced lasalocid production in different strains is due to *lodAB* versus *lodC* and to uncover their repression mechanism in lasalocid biosynthesis.

CSRs are often pathway-specific regulators that function as switches controlling the onset of antibiotic production (32). We used the consensus binding sequences found for LodR1 and LodR2 as queries for a local BLAST search against the genomes of *Streptomyces* sp. FXJ1.172 and *S. lasalocidi* ATCC 31180[T], and we identified no other homologs. Thus, these regulators may not bind to the promoter regions of other genes, indicating that they tend to function as pathway-specific regulators instead of pleiotropic regulators. Nevertheless, antibiotics as well as their intermediates from the biosynthetic pathways can also modulate the activity of their cognate CSRs (35). This generates feedback regulation of antibiotic biosynthesis, which is important for self-protection against the overproduction of antibiotics (32, 36, 37). In this study, we demonstrate that lasalocid specifically binds to the MarR family transcription repressor LodR2 and causes its dissociation from the promoter region of *lodR2–lodE* (Fig. 5B), thus activating the transcription of *lodE*. This finding coincides with the defining feature of MarR family regulators: they normally respond to specific ligands, resulting in attenuated DNA-binding activity and often the derepression of target gene transcription (38). Recently, a polyether antibiotic, calcimycin, has also been proven to inhibit the DNA-binding activity of its cognate CSR CalR3, which subsequently initiates the expression of the calcimycin transporter-encoding gene *calT* (39). Our results further verify that polyether antibiotics can act as modulating ligands of their cognate CSRs, and more polyether ionophores should be used to test the ligand-binding specificity of LodR2 in future studies. The transporter LodE is also crucial for lasalocid biosynthesis because disruption of *lodE* results in reduced lasalocid production and an increased ratio of intracellular/extracellular lasalocid concentrations. It is noteworthy that the residual production of lasalocid in the Δ*lodE* mutant implies the existence of other efflux systems in the producer strain, as seen in the salinomycin producer *Streptomyces albus* BK3-25 (40). Intriguingly, the Δ*lodE* mutant has an intracellular level of lasalocid equivalent to that of FXJ1.172-WT at day 4 but exhibits a greatly reduced transcriptional profile (Fig. 3B and Fig. 5F). A possible explanation for these differences observed is as follows. Lasalocid acts as an ionophore antibiotic, which facilitates cation transport across plasma membranes. Although the intracellular concentration of the *lodE* mutant is similar to that of FXJ1.172-WT at day 4, there are still significant differences between the intracellular and extracellular lasalocid concentrations in the Δ*lodE* mutant. This may lead to the imbalance of ions across the membrane and subsequent cell damage. The cells may sense the changes in ion concentrations and thus downregulate the expression of *lod* to

avoid self-damage. Additional transcriptome analysis will help to explain this confusing phenomenon that we observed.

The distribution of highly similar homologs of *lodR2–lodE* in the other reported lasalocid BGCs highlights their indispensable role in the regulation of lasalocid biosynthesis. Furthermore, *lodE* homologs are also present in other polyether ionophore BGCs, i.e., *idmC*, *monT*, and *ladT* (with 67 to 69% nucleotide sequence identities to *lodE*) from the indanomycin, monensin, and laidlomycin (a monensin analog) gene clusters, respectively (31, 41, 42). Analogous to *lodE* coupled with *lodR2*, each of these homologous genes is flanked by a gene encoding a TetR/MarR family regulator (see Fig. S5 in the supplemental material). This observation indicates that similar regulatory systems may be widespread in different polyether producers. Hence, the disruption of these TetR/MarR family regulator-encoding genes or the overexpression of *lodE* homologs is potentially applicable for generating valuable polyether-overproducing mutants.

The large ATP-binding regulators of the LuxR family (LALs) often serve as CSRs in secondary metabolite biosynthesis in *Streptomyces* (32). Recently, a LAL member named SlnR was identified as a positive pathway-specific regulator in *Streptomyces albus* for the biosynthesis of salinomycin, another marketed carboxylic polyether ionophore (11). Akin to how SlnR modulates salinomycin biosynthesis by activating the transcription of most of the structural genes, LodR3 is essential for the transcription of genes responsible for the formation of the lasalocid skeleton (Fig. 6B). Based on the results of *gusA* assays and cotranscriptional analyses, *lodR3* may be dually controlled by itself and *lodR2*. The high sequence identity between *lodR3* and *las4* suggests the conserved role of this regulatory system in lasalocid biosynthesis, which is confirmed by the results of cross-complementation (Fig. 7B). Noticeably, both *lodR3* and *las4* contain rare TTA codons, suggesting that their expression may rely on the *Streptomyces* development-associated gene *bldA*, which encodes a tRNA for a rare leucine codon, UUA (43).

Taken together, we propose a plausible regulatory model for lasalocid biosynthesis (Fig. 8). In the early stage of *Streptomyces* sp. FXJ1.172 growth, LodR2 specifically binds to the bidirectional promoter region between *lodR2* and *lodE*, inhibiting the transcription of these two genes and part of *lodR3*. Meanwhile, *lodR3* has a separate promoter of its own, and LodR3 can initiate the transcription of most structural genes, leading to the production of lasalocid. When lasalocid accumulates and reaches a certain threshold in the cells, its allosteric interaction with LodR2 induces a conformational change, causing the dissociation of LodR2 from the *lodE* promoter region. Subsequently, the strong expression of *lodE*, which encodes an antibiotic efflux pump, results in a drop in the intracellular lasalocid concentration and then the continuous production of lasalocid. Thus, with the LodR2–LodE system, the concentration of lasalocid in the producing strain is controlled at an appropriate level to avoid self-inhibition. Moreover, LodR1 positively regulates lasalocid biosynthesis by repressing the expression of adjacent *lodA–lodC*, which seems to serve as an alternative regulatory system in addition to the conserved LodR2-LodE and LodR3 systems. In addition, a previously neglected gene, *lodQ* (homologous to *las7*), was unexpectedly identified as a putative target gene of LodR3 in *gusA* transcriptional fusion assays (Fig. 6B). *lodQ* (*las7*) encodes a hypothetical protein with no proposed function in lasalocid biosynthesis. However, the disruption of *las7* led to a remarkable decrease in lasalocid production in *S. lasalocidi* (Minghao Liu, Kairui Wang, and Ying Huang, unpublished data), implying an important role of *lodQ/las7* in lasalocid biosynthesis. Thus, the detailed function of *lodQ/las7* deserves further investigation. Based on our results, multiple genetic modifications can be performed to further improve lasalocid yields. These may include the overexpression of *odR3* (*las4* or *lsd8*), *lodE* (*las2* or *lsd5*), and *dasR*-like positive pleiotropic regulatory genes in a Δ*lodR2* (Δ*las3* or Δ*lsd6–7*) mutant.

## MATERIALS AND METHODS

**Bacterial strains, plasmids, and growth conditions.** The strains and plasmids used in this study are listed in Tables S1 and S2, respectively, in the supplemental material. To harvest *Streptomyces* spores, *Streptomyces* sp. FXJ1.172 and its derivatives, *S. lasalocidi* ATCC 31180[T] and its derivatives, and

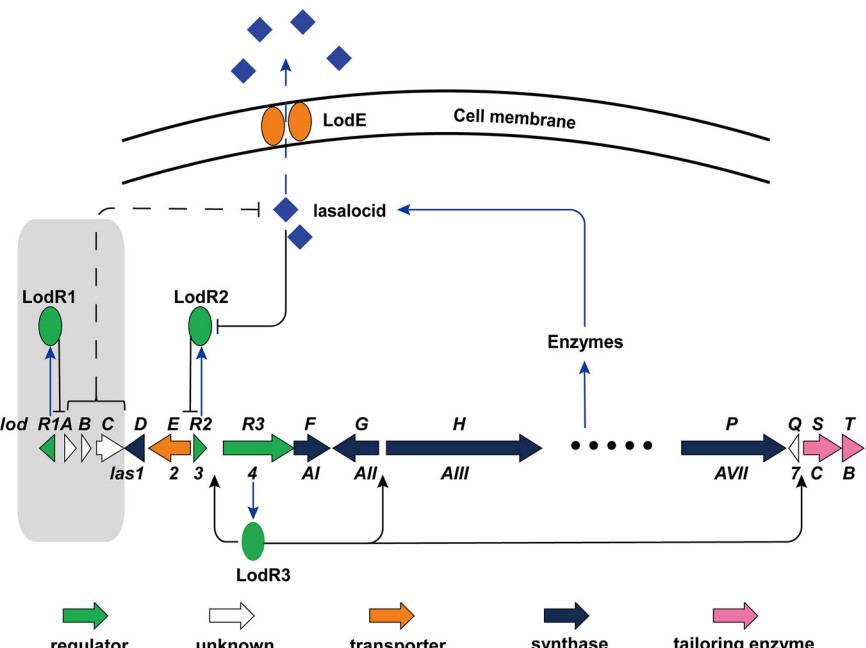

**FIG 8** Putative regulatory model for lasalocid production in *Streptomyces*. The *lodR1–lodC* locus found exclusively in *Streptomyces* sp. FXJ1.172 is highlighted in a light-gray background. Black arrows, activation; black bars, repression; dashed line, the mechanism of repressing lasalocid production needs to be further clarified.

*S. coelicolor* M1146 were grown at 28°C on International *Streptomyces* Project (ISP) medium 3 agar (44), glucose-yeast extract-malt extract (GYM) agar (25), and MS (2% mannitol, 2% soya flour) agar, respectively. *E. coli* strains for nucleic acid manipulation were cultured at 37°C in Luria-Bertani (LB) medium. When necessary, antibiotics were added at the following final concentrations: 100 μg/mL ampicillin, 50 μg/mL apramycin or kanamycin, and 25 μg/mL chloramphenicol for *E. coli* and 25 μg/mL apramycin, kanamycin, or nalidixic acid for *Streptomyces*.

**Construction of gene disruption and complementation mutants.** Nucleic acid manipulations were performed using standard procedures for *E. coli* (45) and *Streptomyces* (46) and according to the instructions of the manufacturers of the restriction enzymes and kits. The lasalocid BGC sequences and genome sequences of *Streptomyces* were obtained from the NCBI GenBank database. The *Streptomyces* mutant strains were constructed from wild-type strains by gene replacement via homologous recombination as previously described (26, 47). For the genetic complementation of *lodR1*, a DNA fragment carrying the *lodR1* ORF was amplified by PCR using primers lodR1-orf-F and lodR1-orf-R (Table S3). The PCR product was gel purified and then ligated with EcoRV-digested pSET152 and a strong constitutive promoter, $P_{kasO}$, by Gibson assembly to give plasmid pSET152::$P_{kasO}$-*lodR1C*. The recombinant plasmid was subsequently introduced into the *lodR1* mutant (Δ*lodR1*) via conjugation to generate the Δ*lodR1*C complemented strain. Other complemented strains and heterologous-expression strains were constructed similarly. The primers used in this study are listed in Table S3.

**Analysis of lasalocid production and bacterial biomass.** For lasalocid production, a suitable number of spores (~$2.5 \times 10^6$) of *Streptomyces* sp. FXJ1.172-WT, wild-type *S. lasalocidi* ATCC 31180$^T$, or their mutants was inoculated into a 50-mL centrifuge tube containing 10 mL of liquid tryptic soy broth (TSB) (pancreatic digest of 1.7% casein and papaic digest of 0.3% soybean, 0.25% dextrose, 0.5% NaCl, and 0.25% $K_2HPO_4$ [pH 7.1 to 7.5]) and incubated on a rotary shaker at 220 rpm at 28°C for 48 h as a seed culture. Next, 1 mL of the seed culture was transferred into 50 mL of liquid SSC (2% soluble starch, 1% soybean powder, 0.5% calcium carbonate) medium. The cultures were incubated for different numbers of days at 28°C before sampling. For the analysis of lasalocid, 50 mL of fermentation broth was centrifuged at 5,000 rpm for 15 min to separate the mycelia and the supernatant. The mycelia were extracted with 50 mL of ethanol, and the supernatant was extracted with an equal volume of ethyl acetate. The extracts were combined and concentrated *in vacuo* to evaporate the solvent, and the residue was redissolved in 1.5 mL of methanol (MeOH) for analysis. HPLC analysis was performed on a Shimadzu Prominence HPLC system using a Waters Xbridge octyl decyl silane column with a linear gradient of MeOH-$H_2O$ at a flow rate of 1.0 mL/min. Mobile phases consisted of A: MeOH; and B: $H_2O$. The following program was used to elute the column: 0 min, 20% A; 15 min, 100% A; 20.5 min, 100% A; 20.51 min, 20% A; 26.5 min 20% A. The effluent was monitored at 316 nm. Tandem mass spectrometry identification of lasalocid was performed as previously described (26). The lasalocid A standard was used as a positive control for HPLC and mass spectrometry analyses to confirm the production of lasalocid (Fig. S6). The bacterial biomass was determined by measuring the cell dry weight. An aliquot of 2 mL of the cell culture was harvested

and centrifuged at 12,000 rpm for 10 min to separate the mycelia and the supernatant, and the mycelia were then dried at 50°C to a constant weight.

**RNA isolation and transcriptional analyses.** Total RNAs were isolated from *Streptomyces* sp. FXJ1.172 and its derivatives after incubation for 72, 96, and 120 h in SSC medium as described previously (48). Likewise, total RNAs were isolated from *S. lasalocidi* ATCC 31180$^T$ and its derivatives after incubation for 120 and 144 h in SSC medium. First-strand synthesis of cDNA was carried out with a Superscript III first-strand synthesis system (Invitrogen, CA, USA) using 1 $\mu$g of total RNA in a total volume of 20 $\mu$L according to the manufacturer's instructions. Cotranscriptional analysis was performed by RT-PCR according to previously reported procedures (49) to ascertain the transcriptional units of *lod* and *las*. Subsequent RT-qPCR of selected genes was carried out in a reaction mixture containing 10 pmol of each primer and 1 $\mu$L of cDNA in a total volume of 20 $\mu$L. RT-qPCR was also conducted without reverse transcriptase to test DNA contamination in the RNA. The 16S rRNA gene was used as an internal control to normalize the RNA concentration. Primers used for transcriptional analyses are listed in Table S3.

**Expression and purification of LodR1-His$_6$, LodR2-His$_6$, and Las3-His$_6$ proteins.** The *lodR1*, *lodR2*, *lodR3*, and *las3* coding regions were amplified using primer pairs lodR1-orf-F/lodR1-orf-R, lodR2-orf-F/lodR2-orf-R, lodR3-orf-F/lodR3-orf-R, and las3-orf-F/las3-orf-R (Table S3), respectively. The amplified fragments were digested with NdeI and XhoI and then inserted into the same sites of pET23b or pET28a to generate the expression plasmids. The recombinant plasmids were confirmed by sequencing and subsequently introduced into *E. coli* BL21(DE3) for protein expression. The transformants were grown in LB medium with 100 $\mu$g/mL ampicillin at 37°C to an optical density at 600 nm (OD$_{600}$) of 0.6. Isopropyl-$\beta$-D-thiogalactopyranoside was then added to a final concentration of 0.1 mM, and the cultures were incubated for an additional 16 h at 28°C. The recombinant proteins were purified using nickel-nitrilotriacetic acid (Ni-NTA) agarose chromatography according to the manufacturer's protocol (Novagen). Protein purity was determined by Coomassie blue staining after SDS-PAGE on a 10% polyacrylamide gel. The concentration of protein was determined according to the protocol of a bicinchoninic acid (BCA) protein assay kit (Novagen). The purified proteins were stored at 4°C.

**EMSAs and DNase I footprinting.** EMSAs were performed mainly as previously described (50). DNA probes covering the intergenic regions were amplified using primer pairs listed in Table S3. The probe (10 ng) was incubated individually with various quantities of the recombinant proteins at 25°C for 25 min in a buffer containing 20 mM Tris-HCl (pH 7.5), 1 mM dithiothreitol (DTT), 10 mM MgCl$_2$, 0.5 $\mu$g/$\mu$L calf bovine serum albumin (BSA), and 5% (vol/vol) glycerol in a total volume of 20 $\mu$L. After incubation, protein-bound and free DNAs were separated by electrophoresis on nondenaturing 4.5% (wt/vol) polyacrylamide gels. Gels were stained with SYBR gold (Invitrogen), and the images were captured by a Bio-Rad GelDoc 2000 system.

DNase I footprinting assays were performed as previously described (51), with slight modifications. To determine the LodR1- and LodR2-binding sites, $^{32}$P-labeled probes P$_{R1-A}$ and P$_{R2-E}$ were amplified by PCR with the labeled primers (Table S3). The footprinting reaction mixture contained 30,000 cpm of the $^{32}$P-labeled DNA probe, 0.2 to 1 $\mu$M LodR1-His$_6$ or LodR2-His$_6$, and 2.5 $\mu$g of poly(dI-dC) (Sigma) in 50 $\mu$L of EMSA buffer. After incubation at 25°C for 25 min, 5.5 $\mu$L RQ1 RNase-free DNase buffer and 0.2 U DNase I were added to the above-described reaction mixture, followed by further incubation at 25°C for 70 s. Next, the reaction was stopped by the addition of 50 $\mu$L of stop solution (20 mM EGTA, pH 8.0) and 100 $\mu$L of phenol-CH$_3$Cl (1:1, vol/vol) to the mixture. DNA fragments in the mixture were precipitated by the addition of an equal volume of isopropanol, and the pellet was washed with 75% (vol/vol) ethanol. The samples were resuspended, denatured, and electrophoretically fractionated according to previously described procedures (51). The sequencing ladder was made using the Silver Sequence DNA sequencing system (Promega, USA). After electrophoresis, the gels were dried and exposed to Kodak X-ray film. To determine the Las3-binding sites on P$_{las2-3}$, the DNA probe was amplified by PCR with 5′-fluorescein amidite (FAM)-labeled primer pairs. Different concentrations of Las3-His$_6$ and 400 ng of the labeled DNA probe were incubated in EMSA buffer for 25 min at 25°C. Next, samples were digested by DNase I (Promega) at 25°C for 45 s, and digestion was terminated with 20 mM EGTA. After phenol-chloroform extraction and ethanol precipitation, the resulting DNA samples were subsequently loaded onto an Applied Biosystems 3730 DNA analyzer. GeneMarker v2.2.0 software was used to analyze the DNA regions protected by Las3.

***gusA* transcriptional fusion assays.** The *gusA* reporter system was constructed as previously described (33). Briefly, the putative promoter region from *lod* was inserted into the upstream region of a promoterless *gusA* gene to generate pIJ10500::P$_x$-*gusA* plasmids (P$_x$ represents the promoters of the *lod* genes tested). The authenticity of each recombinant plasmid was confirmed by subsequent sequencing. The recombinant plasmid was introduced into *S. coelicolor* M1146 via *E. coli*-*Streptomyces* conjugal transfer and integrated into the *Streptomyces* chromosome at the ΦBT1 *attB* site. The resultant *S. coelicolor* M1146 derivative strains were used as negative controls in $\beta$-glucuronidase (GUS) assays and as conjugation recipients for the overexpression of *lodR3*. Subsequently, pSET152::P$_{kasO^*}$-*lodR3C* containing *lodR3* driven by P$_{kasO^*}$ was introduced into the M1146 derivatives in which *gusA* was controlled by different promoters of *lod* genes. The resultant strains were cultured on AS-1 (0.1% yeast extract, 0.5% soluble starch, 0.25% NaCl, 1% Na$_2$SO$_4$, 0.02% L-Ala, 0.02% L-Arg, 0.05% L-Asn) agar with 5-bromo-4-chloro-3-indolyl-$\beta$-D-glucuronide (X-Gluc) at a final concentration of 100 $\mu$M at 28°C for 72 h to observe the chromogenic reaction.

**Statistical analysis.** For each gene disruption, at least three individual mutants were tested. Other experiments were performed at least in biological triplicate unless otherwise stated, and the data are presented as means $\pm$ standard deviations (SD). Statistical significance was analyzed by unpaired two-tailed Student's *t* test. *P* values of <0.05 were considered statistically significant.

## SUPPLEMENTAL MATERIAL

Supplemental material is available online only.

**SUPPLEMENTAL FILE 1**, PDF file, 0.5 MB.

## ACKNOWLEDGMENTS

This work was supported by grants from the China Manned Space Engineering Program (CMSE) (YYWT-0801-EXP-08) and the National Natural Science Foundation of China (31700008 and 81773615). The funders had no role in the study design, data collection and interpretation, or the decision to submit the work for publication.

We declare that we have no conflict of interest.

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
