## [Reviewer comments · Microbiology Spectrum]

Microbiology Spectrum

Comparative and functional analyses reveal conserved and variable regulatory systems that control lasalocid biosynthesis in different *Streptomyces* species

Minghao Liu, Kairui Wang, Junhong Wei, Ning Liu, Guoqing Niu, Huarong Tan, and Ying Huang

Corresponding Author(s): Ying Huang, Institute of Microbiology Chinese Academy of Sciences

Review Timeline:

Submission Date:	September 20, 2022
Editorial Decision:	November 8, 2022
Revision Received:	January 10, 2023
Editorial Decision:	January 28, 2023
Revision Received:	January 29, 2023
Accepted:	January 31, 2023

Editor: Xiaoyu Tang

Reviewer(s): The reviewers have opted to remain anonymous.

Transaction Report:

DOI: <https://doi.org/10.1128/spectrum.03852-22>

November 8, 2022

Prof. Ying Huang
Institute of Microbiology, Chinese Academy of Sciences
State Key Laboratory of Microbial Resources
NO.1 West Beichen Road, Chaoyang District
Beijing 100101
China

Re: Spectrum03852-22 (Comparative and functional analyses reveal conserved and variable regulatory systems that control lasalocid biosynthesis in different *Streptomyces* species)

Dear Prof. Ying Huang:

Link Not Available

Sincerely,

Xiaoyu Tang

Journals Department
Reviewer comments:

Reviewer #1 (Comments for the Author):

In a well-written manuscript, Lui and colleagues set out to define the pathway-specific regulatory network governing the production of the polyether ionophore lasalocid by multiple *Streptomyces* species. Using RT-PCR, they first established transcriptional units within the lasalocid biosynthetic cluster. They next evaluated the contribution of multiple regulators within this cluster using mutational analyses, coupled with lasalocid production levels, RT-PCR of select genes within each operon, and EMSAs/footprinting to define regulator binding sites.

The work that is presented is largely well done (with a few caveats around choices of controls for some of the experiments), but there are some questions raised about the interpretations of the results, and the lack of any comprehensive model to bring these regulators together. The authors propose that the three main regulators function independently, but the data presented could also indicate that there are regulatory connections between them, and it is important that these connections be considered, and be either validated or refuted by additional experiments.

Specific comments:

- 1) Lines 155-157: It is important to compare the complemented mutant strains to a wild type strain bearing the same integrating plasmid vector as a control (as plasmid integration can change the phenotype of *Streptomyces* strains).
- 2) Figure 3, line 732, and Figure S3, lines 306-307: it is possible that there are additional inter-operonic promoters within the biosynthetic clusters.
- 3) 16S is not a great reference for RT-PCR, given its abundance relative to mRNAs.
- 4) It is not always clear what is being presented with the y-axes of the HPLC analyses figures. Should the baseline levels all be the same, and they are just shown stacked on top of each other for ease of visualization? If this is the case, what does the y-axis in each case represent? Why not have a dedicated y-axis for each small graph?

Because of this y-axis confusion, it is hard to understand the significance of overexpressing KABC in the 31180 strain, as it looks to be producing far less lasalocid than the wild type control or strain expressing R1ABC, yet this is not consistent with what is being reported in lines 210-213. Also, it isn't clear how much of this apparent overproduction is due to *lodC* vs *lodAB*, as strain 31180 seems to have at least a part of *lodC* encoded within its lasalocid biosynthetic cluster. The lack of effect when expressing R1ABC could be because these genes are not well expressed in the heterologous host, and does not necessarily support the proposal that these genes can collectively act in different lasalocid hosts.

5) Line 230-232: The conclusion that R2 and R3 have their own promoters, and the effect of R2 on the expression of other genes in the lasalocid cluster is due to changes in R3 expression and not any direct activity by R2 is possible, but other possibilities have not been ruled out at this stage. Consequently, this conclusion is premature. Similarly, in lines 308-309, it is not clear why the described results lead to the conclusion that *las4* has its own promoter. Could it not also be co-transcribed with *las3/R2*, and *Las3/R2* autoregulate operon expression?

6) How were the transcription start sites determined in figures 4 and 5? Is this a prediction based on sequence gazing, or something that has been experimentally confirmed? This would be useful information to include in the figure legend.

7) Were any other promoter regions tested for R2 binding, or just the *lodE-R2* promoter region? What about the promoter for *lodR1*, as the expression of this gene is also increased (like *lode*) in an R2 mutant?

8) Line 247: 'Activating' lasalocid biosynthesis seems to imply a regulatory function. The RT-PCR data presented in Figure 3B suggest that functional *LodE* is indeed needed for expression of most of the biosynthetic genes for lasalocid; however, it is not clear how this is achieved. In the model proposed in the current discussion, *lodE* is repressed by R2, whose DNA binding activity is relieved in the presence of lasalocid. Once lasalocid levels in the cell start to rise, R2's repression is relieved, with *LodE* levels then rising and export of lasalocid occurring. Given that in a *lodE* mutant, the intracellular levels of lasalocid are equivalent to wild type at day 4, presumably this would mean that R2 DNA binding activity should be relieved like in wild type - but the expression level of everything but *lodA* and *lodC* is significantly reduced relative to wild type. Some additional discussion about what is going on here would be very helpful.

9) The chromogenic assay used in Fig. 6 to test the binding of R3 is nice and suggest that R3 may contribute to the expression of the promoter regions tested, but they cannot exclude other possibilities (R3 activates something in *S. coelicolor* that leads to promoter activation). The authors could try ChIP-qPCR, where R3 is tagged with e.g. FLAG tag, and the protein and associated DNA sequences can be directly pulled out from the cell.

10) The high yield strain involving deletion of *las3* and overexpression of *las4* is impressive! Would an even greater yield be obtained with overexpression of *las2* in this same strain?

11) I'm not sure that the conclusion that the three regulators act independently is supported by the data presented. Is anything known about the control of *lodR1* by R2? And it seems that R2 also controls the expression of R3 via a feed-forward type of mechanism, with R3 then activating the expression of most of the structural biosynthetic genes. A final model figure would also be very useful to include.

Editorial comments:

- 1) Lines 31-33: Do *LodR1* and *LodR2* bind the same sequence? If *LodR1* is an activator, why would its binding lead to gene

repression? Please consider rephrasing this to encompass all LodR1 activities.

2) What is meant by a 'variable' regulator? Not-conserved? Some additional explanation would be helpful for the reader.

3) Line 106-108: please consider rephrasing, as this sentence is currently challenging to understand.

4) Line 172: There also isn't any experimental evidence provided to support the proposal that lodR1, lodC, lodG, and lodQ are all transcribed independently of other genes. Please adjust the language in this section accordingly.

5) Lines 193-195: Why were these particular antibiotics selected for testing as potential ligands for LodR1? A brief explanation for these choices would be helpful to readers.

6) Figure 4C: what do the long lines above and below a majority of the intergenic sequence represent? The sequence protected in the footprinting experiments? Please include this information in the figure legend.

7) Line 202-203: while it is very likely that LodR1 is responsible for repressing the expression of the lodA operon, the experiments presented do not exclude the possible action of other regulators, so it would be worth softening the 'proves' descriptor a bit.

8) Line 255: it is not clear what 'interrupted by mis-annotation' means? Please clarify.

9) Line 267-268: not a sentence - please rephrase.

Reviewer #2 (Comments for the Author):

The manuscript "Comparative and functional analyses reveal conserved and variable regulatory systems that control lasalocid biosynthesis in different *Streptomyces* species" by Liu and co-workers describes the functional analysis of the regulatory genes lodR1-lodR3 of the lasalocid biosynthetic gene cluster from *Streptomyces* sp. FXJ1.172 and the corresponding regulators las3 and las4 from *Streptomyces lasalocidi* ATCC 31180. In addition, the role of the transporters lodE and las2 was investigated.

Overall, this is a very comprehensive and nice piece of work. For lodR1 and lodR3, the authors could demonstrate their positive regulatory function whereas lodR2 seems to negatively regulate lasalocid production. The authors first analyzed the operon organization, then generated, analyzed and complemented corresponding mutants, and finally used qPCR, EMSA and footprinting experiments (GusA fusion assays in case of LodR3) to finally shed light into the independent regulation of the lasalocid BGC. The same set of experiments was used to investigate regulation of lasalocid in *Streptomyces lasalocidi* ATCC 31180.

The manuscript is well written, and the data presented fully support the conclusions made by the authors. The findings are new and relevant for the natural product community.

For the entire manuscript:

If names of gene deletions are given, the gene names should always be in italic, e.g. for Δ lodR1, lodR1 should be in italic.

Please also indicate, how many individual mutants were tested. In the method section, it is stated that triplicates were used, does this mean biological or experimental triplicates. Please clarify.

As consensus sequences were found for the regulators, can additional sequences be found in the genome of the producer strain? Maybe there is a possibility, that these regulators may also bind to other genes, e.g. required for precursor supply. Please clarify.

Line 174-176: The authors state, that there is a possibility that lodR3 may have its own promoter due to a long intergenic region. However, this would be true for other intergenic regions >200bp as well, would it? The fact that intergenic regions can be amplified by RT-PCR does not exclude the presence of an additional promoter. If only the intergenic region upstream of lodR3 can possibly harbor such additional promoter, this sentence should be phrased more precisely.

Line 184-186: The authors state, that Figure 4A shows EMSAs with all potential promoters. However, figure 4A only shows the EMSA results for promoter lodR1-lodA and hrdB as negative control. The EMSAs with LodR1 and the other promoters should be presented in the supplemental file and the sentence should be rephrased accordingly.

Line 206: I would replace the term "cassette" by "locus" or "set of genes". Cassette sometimes implies a recombinant piece of DNA used for gene inactivation's or gene replacements.

Line 260: The findings, that a lodE mutant still produces some lasalocids may also be explained by other efflux systems of the producer strain. This possibility should be mentioned.

Line 468: Please indicate the flow rate used for HPLC analysis.

Figure 1: Streptomyces sp 1.172 should be Streptomyces sp. FXJ1.172

Staff Comments:

Preparing Revision Guidelines

Please return the manuscript within 60 days; if you cannot complete the modification within this time period, please contact me. If you do not wish to modify the manuscript and prefer to submit it to another journal, please notify me of your decision immediately so that the manuscript may be formally withdrawn from consideration by Microbiology Spectrum.

Reviewer comments:

Reviewer #1 (Comments for the Author):

In a well-written manuscript, Liu and colleagues set out to define the pathway-specific regulatory network governing the production of the polyether ionophore lasalocid by multiple *Streptomyces* species. Using RT-PCR, they first established transcriptional units within the lasalocid biosynthetic cluster. They next evaluated the contribution of multiple regulators within this cluster using mutational analyses, coupled with lasalocid production levels, RT-PCR of select genes within each operon, and EMSAs/footprinting to define regulator binding sites.

The work that is presented is largely well done (with a few caveats around choices of controls for some of the experiments), but there are some questions raised about the interpretations of the results, and the lack of any comprehensive model to bring these regulators together. The authors propose that the three main regulators function independently, but the data presented could also indicate that there are regulatory connection between them, and it is important that these connections be considered, and be either validated or refuted by additional experiments.

Answer:

Thank you very much for the helpful and impressive comments. Accordingly, we have rephrased our expressions to make them clear and accurate. Meanwhile, we have also added a summary of the comprehensive regulatory model as well as additional results needed. Please see the answers below to your specific comments.

Specific comments:

1) Lines 155-157: It is important to compare the complemented mutant strains to a wild type strain bearing the same integrating plasmid vector as a control (as plasmid integration can change the phenotype of *Streptomyces* strains).

Answer:

Thanks for your reminder. We have supplemented the HPLC profiling of a wild type strain bearing the same integrating plasmid vector as a control. Please see Figure 2.

2) Figure 3, line 732, and Figure S3, lines 306-307: it is possible that there are additional inter-operonic promoters within the biosynthetic clusters.

Answer:

We assume that you mean additional intra-operonic promoters within the biosynthetic clusters. We agree that co-transcriptional analysis may only be meaningful for inter-operonic promoters. Accordingly, "inter-operonic" was added in the legend of Fig. 3A and the arrow for the intra-operonic promoter of *lodR2–lodR3* was removed from the figure.

3) 16S is not a great reference for RT-PCR, given its abundance relative to mRNAs.

Answer:

To solve this problem for 16S rRNA, the cDNA was diluted appropriately to fit its CT value into a reasonable range and then used for RT-qPCR analysis(1).

4) It is not always clear what is being presented with the y-axes of the HPLC analyses figures. Should the baseline levels all be the same, and they are just shown on top of each other for ease of visualization? If this is the case, what does the y-axis in each case represent? Why not have a dedicated y-axis for each small graph?

Answer:

Sorry for this confusing y-axis. According to your suggestion, we have replaced the common y-axis with dedicated ones and added "UV absorbance intensity" on y-axis. Yes, the baselines are all the same and they are just shown on top of each other for ease of visualization. Please see the updated HPLC analyses figures Figs. 2, 4D and 5E.

Because of this y- stacked axis confusion, it is hard to understand the significance of overexpressing KABC in the 31180 strain, as it looks to be producing far less lasalocid than the wild type control or strain expressing R1ABC, yet this is not consistent with what is being reported in lines 210-213. Also, it isn't clear how much of this apparent overproduction is due to *lodC* vs *lodAB*, as strain 31180 seems to have at least a part of *lodC* encoded within its lasalocid biosynthetic cluster. The lack of effect when expressing R1ABC could be because these genes are not well expressed in the heterologous host, and does not necessarily support the proposal that these genes can collectively act in different lasalocid hosts.

Answer:

Compared to 1.172WT, a substantial loss of lasalocid production and overexpression of *lodABC* were detected in $\Delta lodR1$. We speculate that this reduced production of lasalocid is likely due to overexpression of *lodABC*. That is why we conducted heterologous expression of *lodR1ABC* and *lodABC*. As expected, the constitutive expression of *lodA-lodC* resulted in a dramatic decrease of lasalocid production in 31180-KABC compared to 31180-WT (Fig. 4D), which is consistent with our speculation.

Indeed, it's hard to tell how much of this reduced production is due to *lodC* vs *lodAB*. Strain ATCC 31180^T has only about 1/3 of *LodC* encoded within its lasalocid biosynthetic cluster, which only keeps a residue of the conserved domain found in *LodC* and thus looks like a truncated and useless protein. In future research, disruption of *lodC* and *lodAB* in $\Delta lodR1$ will provide useful clues for this tricky question.

In contrast, lasalocid production in 31180-R1ABC was comparable to that in the WT strain,

which may attribute to the inhibition of *lodA-lodC* expression by LodR1, as seen in FXJ1.172-WT. Although we can't exclude the situation you proposed, i.e., these genes are not well expressed in the heterologous host, our results indicate that the overexpression of *lodA-lodC* results in a reduced production of lasalocid in different strains. We have rephrased our interpretations of the results accordingly (lines 201-205 and 340-345).

5) Line 230-232: The conclusion that R2 and R3 have their own promoters, and the effect of R2 on the expression of other genes in the lasalocid cluster is due to changes in R3 expression and not any direct activity by R2 is possible, but other possibilities have not been ruled out at this stage. Consequently, this conclusion is premature. Similarly, in lines 308-309, it is not clear why the described results lead to the conclusion that *las4* has its own promoter. Could it not also be co-transcribed with *las3/R2*, and *Las3/R2* autoregulate operon expression?

Answer:

We agree that this conclusion is premature here. We have transferred this conclusion into the part of *gusA* assays (lines 271-273) and deleted the similar conclusion in line 298.

6) How were the transcription start sites determined in figures 4 and 5? Is this a prediction based on sequence gazing, or something that has been experimentally confirmed? This would be useful information to include in the figure legend.

Answer:

The transcription start sites were predicted by software. We have corrected the word "predicated" to "predicted" in the figure legend.

7) Were any other promoter regions tested for R2 binding, or just the *lodE-R2* promoter region? What about the promoter for *lodR1*, as the expression of this gene is also increased (like *lode*) in an R2 mutant?

Answer:

As described in the legends of Figs. 4 and 5, all potential inter-operonic promoters indicated in Fig. 3A and the intergenic region of *lodR2-lodR3* were used for EMSAs in this study. No other binding sites were detected for LodR1 or LodR2. Additional figures for EMSAs have been added in supplementary Fig. S3.

8) Line 247: 'Activating' lasalocid biosynthesis seems to imply a regulatory function. The RT-PCR data presented in Figure 3B suggest that functional LodE is indeed needed for expression of most of the biosynthetic genes for lasalocid; however, it is not clear how this is achieved. In the model proposed in the current discussion, *lodE* is repressed by R2, whose DNA binding activity is relieved in the presence of lasalocid. Once lasalocid levels in the cell start to rise, R2's repression is relieved, with LodE levels then rising and export of lasalocid occurring. Given that in a *lodE* mutant, the intracellular levels of lasalocid are equivalent to wild

type at day 4, presumably this would mean that R2 DNA binding activity should be relieved like in wild type - but the expression level of everything but *lodA* and *lodC* is significantly reduced relative to wild type. Some additional discussion about what is going on here would be very helpful.

Answer:

Lasalocid is an ionophore agent which can chelate metal cations to form lipid soluble complexes. These complexes facilitate cation transport across plasma membranes, resulting in membrane depolarization and subsequent cell damage or death. Although the intracellular lasalocid concentration of the *lodE* mutant is similar to that of the wild type strain at day 4, there are still significant differences between the intracellular and extracellular concentrations of lasalocid in the *lodE* mutant. This may lead to the imbalance of ions across the membrane and subsequent cell damage. The cells may sense the change of ion concentrations and thus down-regulate the expression of *lod* to avoid self-damage. Additional transcriptome analysis will help to explain this confusing phenomenon we observed. We have added this in DISCUSSION accordingly (lines 381-391).

9) The chromogenic assay used in Fig. 6 to test the binding of R3 is nice and suggest that R3 may contribute to the expression of the promoter regions tested, but they cannot exclude other possibilities (R3 activates something in *S. coelicolor* that leads to promoter activation). The authors could try ChIP-qPCR, where R3 is tagged with e.g. FLAG tag, and the protein and associated DNA sequences can be directly pulled out from the cell.

Answer:

Thanks for this helpful suggestion. We have tried the ChIP-qPCR. Unfortunately, this experiment remains challenging to us, and we may also try other alternative methods in the future.

10) The high yield strain involving deletion of *las3* and overexpression of *las4* is impressive! Would an even greater yield be obtained with overexpression of *las2* in this same strain?

Answer:

Thanks for your helpful advice. We also plan to do overexpression of *lodE/las2/lsd5* in the corresponding strain, please see lines 440-441.

11) I'm not sure that the conclusion that the three regulators act independently is supported by the data presented. Is anything known about the control of *lodR1* by R2? And it seems that R2 also controls the expression of R3 via a feed-forward type of mechanism, with R3 then activating the expression of most of the structural biosynthetic genes. A final model figure would also be very useful to include.

Answer:

According to your suggestion, we have added a putative regulatory model figure and give a

summary about the regulation in lasalocid biosynthesis. Please see Fig. 8 and lines 830-833.

Is anything known about the control of *lodR1* by R2?

Answer:

Not yet. EMSAs of *Lod R2* showed no specific binding activity to the intergenic region of *lodR1-lodA*. Concerning the enhanced expression of *lodR1* in the *lodR2* mutant, more work is needed to give a satisfactory reply to this question.

And it seems that R2 also controls the expression of R3 via a feed-forward type of mechanism, with R3 then activating the expression of most of the structural biosynthetic genes.

Answer:

An impressive comment! Despite its own putative promoter found for *lodR3* in *gusA* assay, we could not exclude the possible control of *lodR2* on the expression of *lodR3* due to their co-transcription observed. Correspondingly, we have rephrased our words to avoid the use of “independently” and added more explanation in DISCUSSION.

Editorial comments:

1) Lines 31-33: Do *LodR1* and *LodR2* bind the same sequence? If *LodR1* is an activator, why would its binding lead to gene repression? Please consider rephrasing this to encompass all *LodR1* activities.

Answer:

Sorry for the misunderstanding. *LodR1* and *LodR2* do not bind the same sequence based on the EMSA and footprinting results. *LodR1* positively regulates lasalocid biosynthesis by repressing the expression of adjacent *lodA-lodC*. We have added this point to encompass all *LodR1* activities and make the description clearer. Please see lines 32-33 and 151.

2) What is meant by a 'variable' regulator? Not-conserved? Some additional explanation would be helpful for the reader.

Answer:

Indeed, we did mean non-conserved. We have supplemented additional explanation to emphasize the exclusive existence of *LodR1* in strain FXJ1.172. Please see line 24.

3) Line 106-108: please consider rephrasing, as this sentence is currently challenging to understand.

Answer:

According to your helpful suggestion, we have rephrased this sentence to make it clearer and more readable (lines 99-101).

4) Line 172: There also isn't any experimental evidence provided to support the proposal that lodR1, lodC, lodG, and lodQ are all transcribed independently of other genes. Please adjust the language in this section accordingly.

Answer:

Thanks for your suggestion. We have replaced the word 'independently' with 'individually' to avoid misunderstanding. Please see line 161.

5) Lines 193-195: Why were these particular antibiotics selected for testing as potential ligands for LodR1? A brief explanation for these choices would be helpful to readers.

Answer:

Lasalocid is its own product from the biosynthetic gene cluster and seems to be the most probable ligand, while other antibiotics were used due to their different structures to check the binding specificity. Please see additional explanation in lines 183-184.

6) Figure 4C: what do the long lines above and below a majority of the intergenic sequence represent? The sequence protected in the footprinting experiments? Please include this information in the figure legend.

Answer:

Thanks. We have included the information in the legend of Fig. 4C according to your advice.

7) Line 202-203: while it is very likely that LodR1 is responsible for repressing the expression of the lodA operon, the experiments presented do not exclude the possible action of other regulators, so it would be worth softening the 'proves' descriptor a bit.

Answer:

Thanks for your advice. We have replaced the word 'prove' with 'suggest' to soften this descriptor in line 203.

8) Line 255: it is not clear what 'interrupted by mis-annotation' means? Please clarify.

Answer:

Sorry for this vague expression. We guess that you mean original line 225, and have added additional explanation to make it clearer in lines 215-216.

9) Line 267-268: not a sentence - please rephrase.

Answer:

Thanks. We have rephrased our words to make them readable (lines 255-257).

Reviewer #2 (Comments for the Author):

The manuscript "Comparative and functional analyses reveal conserved and variable

regulatory systems that control lasalocid biosynthesis in different *Streptomyces* species" by Liu and co-workers describes the functional analysis of the regulatory genes lodR1-lodR3 of the lasalocid biosynthetic gene cluster from *Streptomyces* sp. FXJ1.172 and the corresponding regulators las3 and las4 from *Streptomyces lasalocidi* ATTC 31180. In addition, the role of the transporters lodE and las2 was investigated.

Overall, this is a very comprehensive and nice piece of work. For lodR1 and lodR3, the authors could demonstrate their positive regulatory function whereas lodR2 seems to negatively regulates lasalocid production. The authors first analyzed the operon organization, then generated, analyzed and complemented corresponding mutants, and finally used qPCR, EMSA and footprinting experiments (GusA fusion assays in case of LodR3) to finally shed light into the independent regulation of the lasalocid BGC. The same set of experiments was used to investigate regulation of lasalocid in *Streptomyces lasalocidi* ATTC 31180.

The manuscript is well written, and the data presented fully support the conclusions made by the authors. The findings are new and relevant for the natural product community.

Answer:

Thank you very much for the positive comments.

For the entire manuscript:

If names of gene deletions are given, the gene names should always be in italic, e.g. for Δ lodR1, lodR1 should be in italic.

Answer:

Thanks for your suggestion, we have changed these names into italic form.

Please also indicate, how many individual mutants were tested. In the method section, it is stated that triplicates were used, does this mean biological or experimental triplicates. Please clarify.

Answer:

For each gene disruption, at least three individual mutants were tested. We have added this in the method section. Other experiments were also performed at least in biological triplicate (lines 572-574).

As consensus sequences were found for the regulators, can additional sequences be found in the genome of the producer strain? Maybe there is a possibility, that these regulators may also bind to other genes, e.g. required for precursor supply. Please clarify.

Answer:

We used the consensus binding sequences as queries for local blast search through the

genomes of the producer strains. No other homologs were identified, and thus these regulators may not bind to other genes. Please see additional explanation in lines 357-361.

Line 174-176: The authors state, that there is a possibility that lodR3 may have its own promoter due to a long intergenic region. However, this would be true for other intergenic regions >200bp as well, would it? The fact that intergenic regions can be amplified by RT-PCR does not exclude the presence of an additional promoter. If only the intergenic region upstream of lodR3 can possibly harbor such additional promoter, this sentence should be phrased more precisely.

Answer:

Many thanks. The other reviewer has a similar opinion. We admit that this conclusion is premature here. We have transferred this conclusion into the part of *gusA* assays and deleted the similar conclusion in original lines 308-309. We have also rephrased the sentence accordingly. Please see lines 271-273.

Line 184-186: The authors state, that Figure 4A shown EMSAs with all potential promoters. However, figure 4A only shows the EMSA results for promoter lodR1-lodA and hrdB as negative control. The EMSAs with LodR1 and the other promoters should be presented in the supplemental file and the sentence should be rephrased accordingly.

Answer:

Thanks for your reminder. We have added additional EMSAs figures in the supplemental file and rephrased the sentence accordingly.

Line 206: I would replace the term "cassette" by "locus" or "set of genes". Cassette sometimes implies a recombinant piece of DNA used for gene inactivation's or gene replacements.

Answer:

We have replaced the term "cassette" by "locus" according to your suggestion.

Line 260: The findings, that a lodE mutant still produces some lasalocids may also be explained by other efflux systems of the producer strain. This possibility should be mentioned.

Answer:

According to your suggestion, we have added this possibility in DISCUSSION (lines 378-381).

Line 468: Please indicate the flow rate used for HPLC analysis.

Answer:

Thanks for your reminder, we have added the flow rate accordingly in line 484.

Figure 1: *Streptomyces* sp 1.172 should be *Streptomyces* sp. FXJ1.172

Answer:

Sorry for our carelessness. We have revised this in Fig. 1.

Reference

1. Kim I, Yang D, Tang X, Carroll JL. 2011. Reference gene validation for qPCR in rat carotid body during postnatal development. BMC Research Notes 4:440.

January 28, 2023

Prof. Ying Huang
Institute of Microbiology Chinese Academy of Sciences
State Key Laboratory of Microbial Resources
NO.1 West Beichen Road, Chaoyang District
Beijing 100101
China

Re: Spectrum03852-22R1 (Comparative and functional analyses reveal conserved and variable regulatory systems that control lasalocid biosynthesis in different *Streptomyces* species)

Dear Prof. Ying Huang:

Thank you for submitting your manuscript to Microbiology Spectrum. As you will see your paper is very close to acceptance. Please modify the manuscript along the lines I have recommended. As these revisions are quite minor, I expect that you should be able to turn in the revised paper in less than 30 days, if not sooner. If your manuscript was reviewed, you will find the reviewers' comments below.

When submitting the revised version of your paper, please provide (1) point-by-point responses to the issues raised by the reviewers as file type "Response to Reviewers," not in your cover letter, and (2) a PDF file that indicates the changes from the original submission (by highlighting or underlining the changes) as file type "Marked Up Manuscript - For Review Only". Please use this link to submit your revised manuscript. Detailed instructions on submitting your revised paper are below.

Link Not Available

Sincerely,

Xiaoyu Tang

Reviewer comments:

Reviewer #1 (Comments for the Author):

The authors have done a terrific job of addressing the comments from the previous reviewers. I had only very minor (largely editorial) comments remaining.

1. The affinity of LodR1 for the R1-A promoter region seems to be much higher in the image presented in Fig. 4A than in Fig. S2A. Do the authors have an explanation for this?

2. Line 24, line 42, etc.: please consider changing the 'variable' descriptor throughout the manuscript to something like 'unique' or 'cluster specific'

3. Line 33-34: Within the abstract, it isn't clear where the lodA-C genes are positioned relative to the two operons described in the previous sentence, and which regulator impacts its expression. To help with reader comprehension, it would be helpful to provide some additional context within this sentence, to connect these genes to the previous sentence.

4. Throughout - it would be useful to be consistent with using either FXJ1.172 or 1.172

5. Line 244: genes/proteins are either homologous or not (there is no 'high homology'). 'High/reasonable protein similarity' would be a better description.

6. When referring to 'co-transcriptional analysis', it would be worth indicating that there are 'at least' 4 transcriptional units (or whatever number was determined here), as may be intra-operonic promoters that haven't been detected by the data presented here.

Reviewer #2 (Comments for the Author):

The revised manuscript "Comparative and functional analyses reveal conserved and variable regulatory systems that control lasalocid biosynthesis in different *Streptomyces* species" by Ying Huang and coworkers has now been modified to incorporate all reviewers suggestions.

Preparing Revision Guidelines

Please return the manuscript within 60 days; if you cannot complete the modification within this time period, please contact me. If you do not wish to modify the manuscript and prefer to submit it to another journal, please notify me of your decision immediately so that the manuscript may be formally withdrawn from consideration by Microbiology Spectrum.

Reviewer comments:

Reviewer #1 (Comments for the Author):

The authors have done a terrific job of addressing the comments from the previous reviewers. I had only very minor (largely editorial) comments remaining.

Answer:

Thank you very much for your patience and time spent on our manuscript. According to your suggestion, we have rephrased our expressions to make them clear and accurate.

1. The affinity of LodR1 for the R1-A promoter region seems to be much higher in the image presented in Fig. 4A than in Fig. S2A. Do the authors have an explanation for this?

Answer:

Sorry for our carelessness. In the first round of EMSAs (Fig. S2A), we did adopt a broad concentration gradient (0, 10, 100, 1000 nM) of LodR1 to find its potential binding probes and non-specific bindings were observed when the proteins were overloaded. Accordingly, we narrowed the concentration gradient (0, 10, 50, 100 nM) and used the newly purified protein to performed the second round of EMSAs (Fig. 4A), which gives a more precise and visible illustration of the affinity of LodR1 for the R1-A promoter region. We also have revised the concentration gradient in the Fig. S2A.

2. Line 24, line 42, etc.: please consider changing the 'variable' descriptor throughout the manuscript to something like 'unique' or 'cluster specific'

Answer:

According to your helpful suggestion, we have appropriately replaced other synonymous expressions with 'variable' to make them internally consistent within the manuscript.

3. Line 33-34: Within the abstract, it isn't clear where the lodA-C genes are positioned relative to the two operons described in the previous sentence, and which regulator impacts its expression. To help with reader comprehension, it would be helpful to provide some additional context within this sentence, to connect these genes to the previous sentence.

Answer:

Thanks for your suggestion, we have modified these sentences to make them clearer and more readable (lines 32-33)

4. Throughout - it would be useful to be consistent with using either FXJ1.172 or 1.172

Answer:

Thanks for your reminder. Accordingly, we have consistently used FXJ1.172 in the manuscript and figures.

5. Line 244: genes/proteins are either homologous or not (there is no 'high homology'). 'High/reasonable protein similarity' would be a better description.

Answer:

Sorry for this misuse. We have revised this according to your suggestion.

6. When referring to 'co-transcriptional analysis', it would be worth indicating that there are 'at least' 4 transcriptional units (or whatever number was determined here), as may be intra-operonic promoters that haven't been detected by the data presented here.

Answer:

Thanks for your helpful advice. We have added 'at least' to make our expression more precise when referring to 'co-transcriptional analysis'. Please see lines 161 and 297.

Reviewer #2 (Comments for the Author):

The revised manuscript "Comparative and functional analyses reveal conserved and variable regulatory systems that control lasalocid biosynthesis in different *Streptomyces* species" by Ying Huang and coworkers has now been modified to incorporate all reviewers suggestions

Answer:

Thanks for your helpful comments and acceptance to our revised version of manuscript.

January 31, 2023

Prof. Ying Huang
Institute of Microbiology Chinese Academy of Sciences
State Key Laboratory of Microbial Resources
NO.1 West Beichen Road, Chaoyang District
Beijing 100101
China

Re: Spectrum03852-22R2 (Comparative and functional analyses reveal conserved and variable regulatory systems that control lasalocid biosynthesis in different *Streptomyces* species)

Dear Prof. Ying Huang:

Your manuscript has been accepted, and I am forwarding it to the ASM Journals Department for publication. You will be notified when your proofs are ready to be viewed.

Sincerely,

Xiaoyu Tang
Editor, Microbiology Spectrum
